# HQ-VAE: Hierarchical Discrete Representation Learning with Variational Bayes

**Yuhta Takida**[†], **Yukara Ikemiya**[†], **Takashi Shibuya**[†], **Kazuki Shimada**[†],
**Woosung Choi**[†], **Chieh-Hsin Lai**[†], **Naoki Murata**[†], **Toshimitsu Uesaka**[†], **Kengo Uchida**[†],
**Liao WeiHsiang**[†], **Yuki Mitsufuji**[‡,‡]
[†]*SonyAI, Tokyo, Japan,* [‡]*Sony Group Corporation, Tokyo, Japan*

**Reviewed on OpenReview:** *https://openreview.net/forum?id=xqAVkqrLjx*

## Abstract

Vector quantization (VQ) is a technique to deterministically learn features with discrete codebook representations. It is commonly performed with a variational autoencoding model, VQ-VAE, which can be further extended to hierarchical structures for making high-fidelity reconstructions. However, such hierarchical extensions of VQ-VAE often suffer from the codebook/layer collapse issue, where the codebook is not efficiently used to express the data, and hence degrades reconstruction accuracy. To mitigate this problem, we propose a novel unified framework to stochastically learn hierarchical discrete representation on the basis of the variational Bayes framework, called hierarchically quantized variational autoencoder (HQ-VAE). HQ-VAE naturally generalizes the hierarchical variants of VQ-VAE, such as VQ-VAE-2 and residual-quantized VAE (RQ-VAE), and provides them with a Bayesian training scheme. Our comprehensive experiments on image datasets show that HQ-VAE enhances codebook usage and improves reconstruction performance. We also validated HQ-VAE in terms of its applicability to a different modality with an audio dataset.

## 1 Introduction

Learning representations with discrete features is one of the core goals in the field of deep learning. Vector quantization (VQ) for approximating continuous features with a set of finite trainable code vectors is a common way to make such representations (Toderici et al., 2016; Theis et al., 2017; Agustsson et al., 2017). It has several applications, including image compression (Williams et al., 2020; Wang et al., 2022) and audio codecs (Zeghidour et al., 2021; Défossez et al., 2022). VQ-based representation methods have been improved with deep generative modeling, especially denoising diffusion probabilistic models (Sohl-Dickstein et al., 2015; Ho et al., 2020; Song et al., 2020; Dhariwal & Nichol, 2021; Hoogeboom et al., 2021; Austin et al., 2021) and autoregressive models (van den Oord et al., 2016; Chen et al., 2018; Child et al., 2019). Learning the discrete features of the target data from finitely many representations enables redundant information to be ignored, and such a lossy compression can be of assistance in training deep generative models on large-scale data. After compression, another deep generative model, which is called a prior model, can be trained on the compressed representation instead of the raw data. This approach has achieved promising results in various tasks, e.g., unconditional generation tasks (Razavi et al., 2019; Dhariwal et al., 2020; Esser et al., 2021b;a; Rombach et al., 2022), text-to-image generation (Ramesh et al., 2021; Gu et al., 2022; Lee et al., 2022a) and textually guided audio generation (Yang et al., 2022; Kreuk et al., 2022). Note that the compression performance of VQ limits the overall generation performance regardless of the performance of the prior model.

Vector quantization is usually achieved with a vector quantized variational autoencoder (VQ-VAE) (van den Oord et al., 2017). Based on the construction of van den Oord et al. (2017), inputs are first encoded and quantized with code vectors, which produces a discrete representation of the encoded feature. The discrete representation is then decoded to the data space to recover the original input. Subsequent developments

incorporated a hierarchical structure into the discrete latent space to achieve high-fidelity reconstructions. In particular, Razavi et al. (2019) developed VQ-VAE into a hierarchical model, called VQ-VAE-2. In this model, multi-resolution discrete latent representations are used to extract local (e.g., texture in images) and global (e.g., shape and geometry of objects in images) information from the target data. Another type of hierarchical discrete representation, called residual quantization (RQ), was proposed to reduce the gap between the feature maps before and after the quantization process (Zeghidour et al., 2021; Lee et al., 2022a).

Despite the successes of VQ-VAE in many tasks, training of its variants is still challenging. It is known that VQ-VAE suffers from codebook collapse, a problem in which most of the codebook elements are not being used at all for the representation (Kaiser et al., 2018; Roy et al., 2018; Takida et al., 2022b). This inefficiency may degrade reconstruction accuracy, and limit applications to downstream tasks. The variants with hierarchical latent representations suffers from the same issue. For example, Dhariwal et al. (2020) reported that it is generally difficult to push information to higher levels in VQ-VAE-2; i.e., codebook collapse often occurs there. Therefore, certain heuristic techniques, such as the exponential moving average (EMA) update (Polyak & Juditsky, 1992) and codebook reset (Dhariwal et al., 2020), are usually implemented to mitigate these problems. Takida et al. (2022b) claimed that the issue is triggered because the training scheme of VQ-VAE does not follow the variational Bayes framework and instead relies on carefully designed heuristics. They proposed stochastically quantized VAE (SQ-VAE), with which the components of VQ-VAE, i.e., the encoder, decoder, and code vectors, are trained within the variational Bayes framework with an SQ operator. The model was shown to improve reconstruction performance by preventing the collapse issue thanks to the *self-annealing* effect (Takida et al., 2022b), where the SQ process gradually tends to a deterministic one during training. We expect that this has the potential to stabilize the training even in a hierarchical model, which may lead to improved reconstruction performance with more efficient codebook usage.

Here, we propose *Hierarchically Quantized VAE* (*HQ-VAE*), a general variational Bayesian model for learning hierarchical discrete latent representations. Figure 1 illustrates the overall architecture of HQ-VAE. The hierarchical structure consists of a *bottom-up* and *top-down* path pair, which helps to capture the local and global information in the data. We instantiate the generic HQ-VAE by introducing two types of *top-down* layer. These two layers formulate hierarchical structures of VQ-VAE-2 and residual-quantized VAE (RQ-VAE) within the variational scheme, which we call *SQ-VAE-2* and *RSQ-VAE*, respectively. HQ-VAE can be viewed as a hierarchical version (extension) of SQ-VAE, and it has the favorable properties of SQ-VAE (e.g., the *self-annealing* effect). In this sense, it unifies the current well-known VQ models in the variational Bayes framework and thus provides a novel training mechanism. We empirically show that HQ-VAE improves upon conventional methods in the vision and audio domains. Moreover, we applied HQ-VAEs to generative tasks on image datasets to show the feasibility of the learnt discrete latent representations. This study is the first attempt at developing variational Bayes on hierarchical discrete representations.

Throughout this paper, uppercase letters ($P$, $Q$) and lowercase letters ($p$, $q$) respectively denote the probability mass functions and probability density functions, calligraphy letters ($\mathcal{P}$, $\mathcal{Q}$) the joint probabilistic distributions of continuous and discrete random variables, and bold lowercase and uppercase letters (e.g., $\boldsymbol{x}$ and $\boldsymbol{Y}$) vectors and matrices. Moreover, the $i$th column vector in $\boldsymbol{Y}$ is written as $\boldsymbol{y}_i$, $[N]$ denotes the set of positive integers no greater than $N \in \mathbb{N}$, and $\mathcal{J}$ and $\mathcal{L}$ denote the objective functions of HQ-VAE and conventional autoencoders, respectively.

## 2 Background

We first review VQ-VAE and its extensions to hierarchical latent models. Then, we revisit SQ-VAE, which serves as the foundation framework of HQ-VAE.

**VQ-VAE.** To discretely represent observations $\boldsymbol{x} \in \mathbb{R}^D$, a codebook $\boldsymbol{B}$ is used that consists of finite trainable code vectors $\{\boldsymbol{b}_k\}_{k=1}^K$ ($\boldsymbol{b}_k \in \mathbb{R}^{d_b}$). A discrete latent variable $\boldsymbol{Z}$ is constructed to be in a $d_z$-tuple of $\boldsymbol{B}$, i.e., $\boldsymbol{Z} \in \boldsymbol{B}^{d_z}$, which is later decoded to generate data samples. A deterministic encoder and decoder pair is used to connect the observation and latent representation, where the encoder maps $\boldsymbol{x}$ to $\boldsymbol{Z}$ and the decoder recovers $\boldsymbol{x}$ from $\boldsymbol{Z}$ by using a decoding function $\boldsymbol{f}_{\boldsymbol{\theta}} : \mathbb{R}^{d_b \times d_z} \to \mathbb{R}^D$. An encoding function, denoted as

$G_{\phi} : \mathbb{R}^D \to \mathbb{R}^{d_b \times d_z}$, and a deterministic quantization operator are used together as the encoder. The encoding function first maps $\boldsymbol{x}$ to $\hat{\boldsymbol{Z}} \in \mathbb{R}^{d_b \times d_z}$; then, the quantization operator finds the nearest neighbor of $\hat{\boldsymbol{z}}_i$ for $i \in [d_z]$, i.e., $\boldsymbol{z}_i = \arg\min_{\boldsymbol{b}_k} \|\hat{\boldsymbol{z}}_i - \boldsymbol{b}_k\|_2^2$. The trainable components (encoder, decoder, and codebook) are learned by minimizing the objective,

$$\mathcal{L}_{\text{VQ-VAE}} = \|\boldsymbol{x} - \boldsymbol{f_\theta}(\boldsymbol{Z})\|_2^2 + \beta \|\hat{\boldsymbol{Z}} - \text{sg}[\boldsymbol{Z}]\|_F^2, \tag{1}$$

where $\text{sg}[\cdot]$ is the stop-gradient operator and $\beta$ is a hyperparameter balancing the two terms. The codebook is updated by applying the EMA update to $\|\text{sg}[\hat{\boldsymbol{Z}}] - \boldsymbol{Z}\|_F^2$.

**VQ-VAE-2.** Razavi et al. (2019) incorporated hierarchical structure into the discrete latent space in VQ-VAE to model local and global information separately. The model consists of multiple levels of latents so that the top levels have global information, while the bottom levels focus on local information, conditioned on the top levels. The training of the model follows the same scheme as the original VQ-VAE (e.g., stop-gradient, EMA update, and deterministic quantization).

**RQ-VAE.** As an extension of VQ, RQ was proposed to provide a finer approximation of $\boldsymbol{Z}$ by taking into account information on quantization gaps (residuals) (Zeghidour et al., 2021; Lee et al., 2022a). With RQ, $L$ code vectors are assigned to each vector $\boldsymbol{z}_i$ ($i \in [d_z]$), instead of increasing the codebook size $K$. To make multiple assignments, RQ repeatedly quantizes the target feature and computes quantization residuals, denoted as $\boldsymbol{R}_l$. Namely, the following procedure is repeated $L$ times, starting with $\boldsymbol{R}_0 = \hat{\boldsymbol{Z}}$: $\boldsymbol{z}_{l,i} = \arg\min_{\boldsymbol{b}_k} \|\boldsymbol{r}_{l-1,i} - \boldsymbol{b}_k\|_2^2$ and $\boldsymbol{R}_l = \boldsymbol{R}_{l-1} - \boldsymbol{Z}_l$. By repeating RQ, the discrete representation can be refined in a coarse-to-fine manner. Finally, RQ discretely approximates the encoded variable as $\hat{\boldsymbol{Z}} \approx \sum_{l=1}^{L} \boldsymbol{Z}_l$, where the conventional VQ is regarded as a special case of RQ with $L = 1$.

**SQ-VAE.** Takida et al. (2022b) proposed a variational Bayes framework for learning the VQ-VAE components to mitigate the issue of codebook collapse. The resulting model, SQ-VAE, also has deterministic encoding/decoding functions and a trainable codebook like the above autoencoders. However, unlike the deterministic quantization schemes of VQ and RQ, SQ-VAE adopts stochastic quantization (SQ) for the encoded features to approximate the categorical posterior distribution $P(\boldsymbol{Z}|\boldsymbol{x})$. More precisely, it defines a stochastic dequantization process $p_{s^2}(\tilde{\boldsymbol{z}}_i|\boldsymbol{Z}) = \mathcal{N}(\tilde{\boldsymbol{z}}_i; \boldsymbol{z}_i, s^2\boldsymbol{I})$, which converts a discrete variable $\boldsymbol{z}_i$ into a continuous one $\tilde{\boldsymbol{z}}_i$ by adding Gaussian noise with a learnable variance $s^2$. By Bayes' rule, this process is associated with the reverse operation, i.e., SQ, which is given by $\hat{P}_{s^2}(\boldsymbol{z}_i = \boldsymbol{b}_k|\tilde{\boldsymbol{Z}}) \propto \exp\left(-\frac{\|\tilde{\boldsymbol{z}}_i - \boldsymbol{b}_k\|_2^2}{2s^2}\right)$. Thanks to this variational framework, the degree of stochasticity in the quantization scheme becomes adaptive. This allows SQ-VAE to benefit from the effect of *self-annealing*, where the SQ process gradually approaches a deterministic one as $s^2$ decreases. This generally improves the efficiency of codebook usage. Other related works in the literature of discrete posterior modeling are found in Appendix A.

## 3 Hierarchically quantized VAE

In this section, we formulate the generic HQ-VAE model, which learns a hierarchical discrete latent representation within the variational Bayes framework. It serves as a backbone of the instances of HQ-VAE presented in Section 4.

To achieve a hierarchical discrete representation of depth $L$, we first introduce $L$ groups of discrete latent variables, which are denoted as $\boldsymbol{Z}_{1:L} := \{\boldsymbol{Z}_l\}_{l=1}^{L}$. For each $l \in [L]$, we further introduce a trainable codebook, $\boldsymbol{B}_l := \{\boldsymbol{b}_k^l\}_{k=1}^{K_l}$, consisting of $K_l$ $d_b$-dimensional code vectors, i.e., $\boldsymbol{b}_k^l \in \mathbb{R}^{d_b}$ for $k \in [K_l]$. The variable $\boldsymbol{Z}_l$ is represented as a $d_l$-tuple of the code vectors in $\boldsymbol{B}_l$; namely, $\boldsymbol{Z}_l \in \boldsymbol{B}_l^{d_l}$. Similarly to conventional VAEs, the latent variable of each group is assumed to follow a pre-defined prior mass function. We set the prior as an i.i.d. uniform distribution, defined as $P(\boldsymbol{z}_{l,i} = \boldsymbol{b}_k) = 1/K_l$ for $i \in [d_l]$. The probabilistic decoder is set to a normal distribution with a trainable isotropic covariance matrix, $p_{\boldsymbol{\theta}}(\boldsymbol{x}|\boldsymbol{Z}_{1:L}) = \mathcal{N}(\boldsymbol{x}; \boldsymbol{f_\theta}(\boldsymbol{Z}_{1:L}), \sigma^2\boldsymbol{I})$ with a decoding function $\boldsymbol{f_\theta} : \mathbb{R}^{d_b \times d_1} \oplus \cdots \oplus \mathbb{R}^{d_b \times d_L} \to \mathbb{R}^D$. To generate instances, it decodes latent variables sampled from the prior. Here, the exact evaluation of $P_{\boldsymbol{\theta}}(\boldsymbol{Z}_{1:L}|\boldsymbol{x})$ is required to train the generative model with the maximum likelihood. However, this is intractable in practice. Thus, we introduce an approximated posterior on $\boldsymbol{Z}_{1:L}$ given $\boldsymbol{x}$ and derive the evidence lower bound (ELBO) for maximization instead.

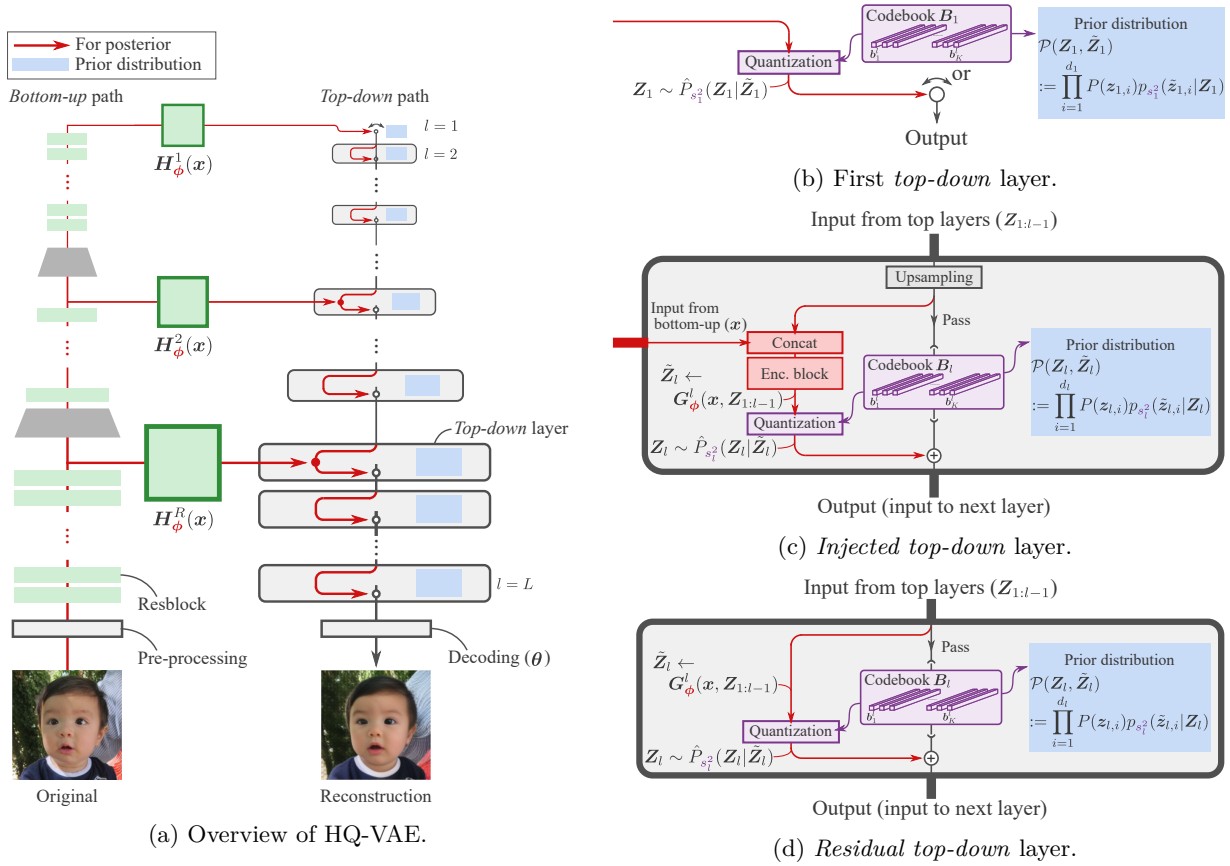

Figure 1: (a) HQ-VAE consists of *bottom-up* and *top-down* paths. Red arrows represent the approximated posterior (Equation (3)). This process consists of the following three steps: the *bottom-up* path generates the features $\{\boldsymbol{H}_\phi^r(\boldsymbol{x})\}_{r=1}^R$ at different resolutions; the *top-down* path fuses each of them with the information processed in the *top-down* path into $\tilde{\boldsymbol{Z}}_{1:L}$; the auxiliary variable at the $l$th layer, $\tilde{\boldsymbol{Z}}_l$, is quantized into $\boldsymbol{Z}_l$ based on the codebook $\boldsymbol{B}_l$. Lastly, to complete the autoencoding process, all the quantized latent variables in the *top-down* path, $\boldsymbol{Z}_{1:L}$, are fed to the decoder $\boldsymbol{f_\theta}$, recovering the original data sample. The objective function takes the Kullback–Leibler divergence of posterior and prior (in the blue box). (b) First layer for *top-down* path. Any HQ-VAE should include this layer as its top layer. An HQ-VAE that includes only this layer in its *top-down* path reduces to an SQ-VAE. (c)-(d) Two types of top-down layers are proposed: *injected top-down* and *residual top-down* layers. *Injected top-down* layer newly injects the higher-resolution features generated by the *bottom-up* path into the latent variables in the previous *top-down* layers, resulting in a higher resolution (i.e., larger dimension) of $\tilde{\boldsymbol{Z}}_l$ and $\boldsymbol{Z}_l$ compared to $\boldsymbol{Z}_{1:l-1}$. *Residual top-down* layer does not incorporate any new features from the *bottom-up* path but aims to refine the processed latent variables in *top-down* path with additional code vectors for representing the quantization residual from the preceding *top-down* layer. As a result, it maintains the same dimensionality for $\boldsymbol{Z}_l$ as in the previous layer (i.e., $\boldsymbol{Z}_{l-1}$). An HQ-VAE that consists entirely of the injected (residual) *top-down* layers is analogous to VQ-VAE-2 (RQ-VAE), which is presented in Section 4.2.1 (4.3.1).

Inspired by the work on hierarchical Gaussian VAEs (Sønderby et al., 2016; Vahdat & Kautz, 2020; Child, 2021), we introduce HQ-VAE *bottom-up* and *top-down* paths, as shown in Figure 1a. The approximated posterior has a *top-down* structure ($\boldsymbol{Z}_1 \to \boldsymbol{Z}_2 \to \cdots \to \boldsymbol{Z}_L$). The *bottom-up* path first generates features from $\boldsymbol{x}$ as $\boldsymbol{H}_\phi^r(\boldsymbol{x})$ at different resolutions ($r \in [R]$). The latent variable in each group on the *top-down* path is processed from $\boldsymbol{Z}_1$ to $\boldsymbol{Z}_L$ in that order by taking $\boldsymbol{H}_\phi^r(\boldsymbol{x})$ into account. To do so, two features, one extracted by the *bottom-up* path ($\boldsymbol{H}_\phi^r(\boldsymbol{x})$) and one processed on the higher layers of the *top-down* path ($\boldsymbol{Z}_{1:l-1}$), can be fed to each layer and used to estimate the $\boldsymbol{Z}_l$ corresponding to $\boldsymbol{x}$, which we define as

$\hat{\boldsymbol{Z}}_l = \boldsymbol{G}_{\boldsymbol{\phi}}^l(\boldsymbol{H}_{\boldsymbol{\phi}}^r(\boldsymbol{x}), \boldsymbol{Z}_{1:l-1})$. The $l$th group $\boldsymbol{Z}_l$ has a unique resolution index $r$; we denote it as $r(l)$. For simplicity, we will ignore $\boldsymbol{H}_{\boldsymbol{\phi}}^r$ in $\hat{\boldsymbol{Z}}_l$ and write $\hat{\boldsymbol{Z}}_l = \boldsymbol{G}_{\boldsymbol{\phi}}^l(\boldsymbol{x}, \boldsymbol{Z}_{1:l-1})$. The design of the encoding function $\boldsymbol{G}_{\boldsymbol{\phi}}^l$ lead us to a different modeling of the approximated posterior. A detailed discussion of it is presented in the next section.

It should be noted that the outputs of $\boldsymbol{G}_{\boldsymbol{\phi}}^l$ lie in $\mathbb{R}^{d_z \times d_b}$, whereas the support of $\boldsymbol{Z}_l$ is restricted to $\boldsymbol{B}_l^{d_z}$. To connect these continuous and discrete spaces, we use a pair of stochastic dequantization and quantization processes, as in Takida et al. (2022b). We define the stochastic dequantization process for each group as $p_{s_l^2}(\tilde{\boldsymbol{z}}_{l,i}|\boldsymbol{Z}_l) = \mathcal{N}(\tilde{\boldsymbol{z}}_{l,i}; \boldsymbol{z}_{l,i}, s_l^2\boldsymbol{I})$, which is equivalent to adding Gaussian noise to the discrete variable the covariance of which, $s_l^2\boldsymbol{I}$, depends on the index of the group $l$. We hereafter denote the set of $\tilde{\boldsymbol{Z}}_l$ as $\tilde{\boldsymbol{Z}}_{1:L}$, i.e., $\tilde{\boldsymbol{Z}}_{1:L} := \{\tilde{\boldsymbol{Z}}_l\}_{l=1}^L$. Next, we derive a stochastic quantization process in the form of the inverse operator of the above stochastic dequantization:

$$\hat{P}_{s_l^2}(\boldsymbol{z}_{l,i} = \boldsymbol{b}_k|\tilde{\boldsymbol{Z}}_l) \propto \exp\left(-\frac{\|\tilde{\boldsymbol{z}}_{l,i} - \boldsymbol{b}_k\|_2^2}{2s_l^2}\right). \tag{2}$$

By using these stochastic dequantization and quantization operators, we can connect $\hat{\boldsymbol{Z}}_{1:L}$ and $\boldsymbol{Z}_{1:L}$ via $\tilde{\boldsymbol{Z}}_{1:L}$ in a stochastic manner, which leads to the entire encoding process:

$$\mathcal{Q}(\boldsymbol{Z}_{1:L}, \tilde{\boldsymbol{Z}}_{1:L}|\boldsymbol{x}) = \prod_{l=1}^{L}\prod_{i=1}^{d_l} p_{s_l^2}(\tilde{\boldsymbol{z}}_{l,i}|\boldsymbol{G}_{\boldsymbol{\phi}}^l(\boldsymbol{x}, \boldsymbol{Z}_{1:l-1}))\hat{P}_{s_l^2}(\boldsymbol{z}_{l,i}|\tilde{\boldsymbol{Z}}_l). \tag{3}$$

The prior distribution on $\boldsymbol{Z}_{1:L}$ and $\tilde{\boldsymbol{Z}}_{1:L}$ is defined using the stochastic dequantization process as

$$\mathcal{P}(\boldsymbol{Z}_{1:L}, \tilde{\boldsymbol{Z}}_{1:L}) = \prod_{l=1}^{L}\prod_{i=1}^{d_l} P(\boldsymbol{z}_{l,i})p_{s_l^2}(\tilde{\boldsymbol{z}}_i|\boldsymbol{Z}_l), \tag{4}$$

where the latent representations are generated from $l = 1$ to $L$ in that order. The generative process does not use $\tilde{\boldsymbol{Z}}$; rather it uses $\boldsymbol{Z}$ as $\boldsymbol{x} = \boldsymbol{f}_{\boldsymbol{\theta}}(\boldsymbol{Z})$.

## 4 Instances of HQ-VAE

Now that we have established the overall framework of HQ-VAE, we will consider two special cases with different *top-down* layers: *injected top-down* or *residual top-down*. We will derive two instances of HQ-VAE that consist only of the *injected top-down* layer or the *residual top-down* layer, which we will call SQ-VAE-2 and RSQ-VAE by analogy to VQ-VAE-2 and RQ-VAE. Specifically, SQ-VAE-2 and RSQ-VAE share the same architectural structure as VQ-VAE-2 and RQ-VAE, respectively, but the differences lie in their training strategies, as discussed in Sections 4.2.2 and 4.3.2. Furthermore, these two layers can be combinatorially used to define a hybrid model of SQ-VAE-2 and RSQ-VAE, as explained in Appendix D. Note that the prior distribution (Equation (4)) is identical across all instances.

### 4.1 First top-down layer

The first *top-down* layer is the top of layer in HQ-VAE. As illustrated in Figure 1b, it takes $\boldsymbol{H}_{\boldsymbol{\phi}}^1(\boldsymbol{x})$ as an input and processes it with SQ. An HQ-VAE composed only of this layer reduces to SQ-VAE.

### 4.2 Injected top-down layer

The *injected top-down* layer for the approximated posterior is shown in Figure 1c. This layer infuses the variable processed on the *top-down* path with higher resolution information from the *bottom-up* path. The $l$th layer takes the feature from the *bottom-up* path ($\boldsymbol{H}_{\boldsymbol{\phi}}^{r(l)}(\boldsymbol{x})$) and the variables from the higher layers in the *top-down* path as inputs. The variable from the higher layers is first upsampled to align it with $\boldsymbol{H}_{\boldsymbol{\phi}}^{r(l)}(\boldsymbol{x})$. These two variables are then concatenated and processed in an encoding block. The above overall process

corresponds to $\hat{\boldsymbol{Z}}_l = \boldsymbol{G}_{\boldsymbol{\phi}}^l(\boldsymbol{x}, \boldsymbol{Z}_{1:l-1})$ in Section 3. The encoded variable $\hat{\boldsymbol{Z}}_l$ is then quantized into $\boldsymbol{Z}_l$ with the codebook $\boldsymbol{B}_l$ through the process described in Equation (2)[1]. Finally, the sum of the variable from the top layers and quantized variable $\boldsymbol{Z}_l$ is passed through to the next layer.

### 4.2.1 SQ-VAE-2

An instance of HQ-VAE that has only the *injected top-down* layers in addition to the first layer reduces to SQ-VAE-2. Note that since the index of resolutions and layers have a one-to-one correspondence in this structure, $r(l) = l$ and $L = R$. As in the usual VAEs, we evaluate the evidence lower bound (ELBO) inequality: $\log p_{\boldsymbol{\theta}}(\boldsymbol{x}) \geq -\mathcal{J}_{\text{SQ-VAE-2}}(\boldsymbol{x}; \boldsymbol{\theta}, \boldsymbol{\phi}, \boldsymbol{s}^2, \mathcal{B})$, where

$$\mathcal{J}_{\text{SQ-VAE-2}}(\boldsymbol{x}; \boldsymbol{\theta}, \boldsymbol{\phi}, \boldsymbol{s}^2, \mathcal{B}) = \mathbb{E}_{\mathcal{Q}(\boldsymbol{Z}_{1:L}, \tilde{\boldsymbol{z}}_{1:L} | \boldsymbol{x})} \left[ -\log p_{\boldsymbol{\theta}}(\boldsymbol{x} | \boldsymbol{Z}_{1:L}) + \log \frac{\mathcal{Q}(\boldsymbol{Z}_{1:L}, \tilde{\boldsymbol{Z}}_{1:L} | \boldsymbol{x})}{\mathcal{P}(\boldsymbol{Z}_{1:L}, \tilde{\boldsymbol{Z}}_{1:L})} \right] \quad (5)$$

is the ELBO, $\boldsymbol{s}^2 := \{s_l^2\}_{l=1}^L$, and $\mathcal{B} := (\boldsymbol{B}_1, \cdots, \boldsymbol{B}_L)$. Hereafter, we will omit the arguments of the objective functions for simplicity. By decomposing $\mathcal{Q}$ and $\mathcal{P}$ and substituting parameterizations for the probabilistic parts, we have

$$\mathcal{J}_{\text{SQ-VAE-2}} = \frac{D}{2} \log \sigma^2 + \mathbb{E}_{\mathcal{Q}(\boldsymbol{Z}_{1:L}, \tilde{\boldsymbol{z}}_{1:L} | \boldsymbol{x})} \left[ \frac{\|\boldsymbol{x} - \boldsymbol{f}_{\boldsymbol{\theta}}(\boldsymbol{Z}_{1:L})\|_2^2}{2\sigma^2} + \sum_{l=1}^L \left( \frac{\|\tilde{\boldsymbol{Z}}_l - \boldsymbol{Z}_l\|_F^2}{2s_l^2} - H(\hat{P}_{s_l^2}(\boldsymbol{Z}_l | \tilde{\boldsymbol{Z}}_l)) \right) \right], \quad (6)$$

where $H(\cdot)$ indicates the entropy of the probability mass function and constant terms have been omitted. The derivation of Equation (6) is in Appendix C. The objective function (6) consists of a reconstruction term and regularization terms for $\boldsymbol{Z}_{1:L}$ and $\tilde{\boldsymbol{Z}}_{1:L}$. The expectation w.r.t. the probability mass function $\hat{P}_{s_l^2}(\boldsymbol{z}_{l,i} = \boldsymbol{b}_k | \tilde{\boldsymbol{Z}}_l)$ can be approximated with the corresponding Gumbel-softmax distribution (Maddison et al., 2017; Jang et al., 2017) in a reparameterizable manner.

### 4.2.2 SQ-VAE-2 vs. VQ-VAE-2

VQ-VAE-2 is composed in a similar fashion to SQ-VAE-2, but is trained with the following objective function:

$$\mathcal{L}_{\text{VQ-VAE-2}} = \|\boldsymbol{x} - \boldsymbol{f}_{\boldsymbol{\theta}}(\boldsymbol{Z}_{1:L})\|_2^2 + \beta \sum_{l=1}^L \|\boldsymbol{G}_{\boldsymbol{\phi}}^l(\boldsymbol{x}, \boldsymbol{Z}_{1:l-1}) - \text{sg}[\boldsymbol{Z}_l]\|_F^2, \quad (7)$$

where the codebooks are updated with the EMA update in the same manner as the original VQ-VAE. The objective function (7), except for the stop gradient operator and EMA update, can be obtained by setting both $s_l^2$ and $\sigma^2$ to infinity while keeping the ratio of the variances as $s_l^2 = \beta^{-1}\sigma^2$ for $l \in [L]$ in Equation (6). In contrast, since all the parameters except $D$ and $L$ in Equation (6) are optimized, the weight of each term is automatically adjusted during training. Furthermore, SQ-VAE-2 is expected to benefit from the *self-annealing* effect, as does the original SQ-VAE (see Section 5.3).

### 4.3 Residual top-down layer

In this subsection, we set $R = 1$ for demonstration purposes (the general case of $R$ is in Appendix D). This means the *bottom-up* and *top-down* paths are connected only at the top layer. In this setup, the *bottom-up* path generates only a single-resolution feature; thus, we denote $\boldsymbol{H}_{\boldsymbol{\phi}}^1(\boldsymbol{x})$ as $\boldsymbol{H}_{\boldsymbol{\phi}}(\boldsymbol{x})$ for notational simplicity in this subsection. We design a *residual top-down* layer for the approximated posterior, as in Figure 1d. This layer is to better approximate the target feature with additional assignments of code vectors per encoded vector. By stacking this procedure $L$ times, the feature is approximated as

$$\boldsymbol{H}_{\boldsymbol{\phi}}(\boldsymbol{x}) \approx \sum_{l=1}^L \boldsymbol{Z}_l. \quad (8)$$

---

[1]We empirically found setting $\tilde{\boldsymbol{Z}}_l$ to $\hat{\boldsymbol{Z}}_l$ instead of sampling $\tilde{\boldsymbol{Z}}_l$ from $p_{\boldsymbol{s}^2}(\tilde{\boldsymbol{z}}_{l,i} | \hat{\boldsymbol{Z}}_l)$ leads to better performance (as reported in Takida et al. (2022b); therefore, we follow the procedure in practice.

Therefore, in this layer, only the information from the higher layers (not the one from the *bottom-up* path) is fed to the layer. It is desired that $\sum_{l'=1}^{l+1} \boldsymbol{Z}_{l'}$ approximate the feature better than $\sum_{l'=1}^{l} \boldsymbol{Z}_{l'}$. On this basis, we let the following residual pass through to the next layer:

$$\boldsymbol{G}_{\boldsymbol{\phi}}^{l}(\boldsymbol{x}, \boldsymbol{Z}_{1:l-1}) = \boldsymbol{H}_{\boldsymbol{\phi}}(\boldsymbol{x}) - \sum_{l'=1}^{l-1} \boldsymbol{Z}_{l'}. \tag{9}$$

### 4.3.1 RSQ-VAE

RSQ-VAE is an instance that has only *residual top-down* layers and the first layer. At this point, by following Equation (5) and omitting constant terms, we could derive the ELBO objective in the same form as Equation (6):

$$\mathcal{J}_{\text{RSQ-VAE}}^{\text{naïve}} = \frac{D}{2} \log \sigma^2 + \mathbb{E}_{\mathcal{Q}(\boldsymbol{Z}_{1:L}, \tilde{\boldsymbol{z}}_{1:L}|\boldsymbol{x})} \left[ \frac{\|\boldsymbol{x} - \boldsymbol{f}_{\boldsymbol{\theta}}(\boldsymbol{Z}_{1:L})\|_2^2}{2\sigma^2} + \sum_{l=1}^{L} \left( \frac{\|\tilde{\boldsymbol{Z}}_l - \boldsymbol{Z}_l\|_F^2}{2s_l^2} - H(\hat{P}_{s_l^2}(\boldsymbol{Z}_l|\tilde{\boldsymbol{Z}}_l)) \right) \right], \tag{10}$$

where the numerator of the third term corresponds to the evaluation of the residuals $\boldsymbol{H}_{\boldsymbol{\phi}}(\boldsymbol{x}) - \sum_{l'=1}^{l} \boldsymbol{Z}_{l'}$ for all $l \in [L]$ with the dequantization process. However, we empirically found that training the model with this ELBO objective above was often unstable. We suspect this is because the objective regularizes $\boldsymbol{Z}_{1:L}$ to make $\sum_{l'=1}^{l} \boldsymbol{Z}_{l'}$ close to the feature for all $l \in [L]$. We hypothesize that the regularization excessively pushes information to the top layers, which may result in layer collapse in the bottom layers. To address the issue, we modify the prior distribution $\mathcal{P}$ to result in weaker latent regularizations. Specifically, we define the prior distribution as a joint distribution of $(\boldsymbol{Z}_{1:L}, \tilde{\boldsymbol{Z}})$ instead of that of $(\boldsymbol{Z}_{1:L}, \tilde{\boldsymbol{Z}}_{1:L})$, where $\tilde{\boldsymbol{Z}} = \sum_{l=1}^{L} \tilde{\boldsymbol{Z}}_l$. From the reproductive property of the Gaussian distribution, a continuous latent variable converted from $\boldsymbol{Z}$ via the stochastic dequantization processes, $\tilde{\boldsymbol{Z}} = \sum_{l=1}^{L} \tilde{\boldsymbol{Z}}_l$, follows a Gaussian distribution:

$$p_{\boldsymbol{s}^2}(\tilde{\boldsymbol{z}}_i|\boldsymbol{Z}) = \mathcal{N}\left( \tilde{\boldsymbol{z}}_i; \sum_{l=1}^{L} \boldsymbol{z}_{l,i}, \left( \sum_{l=1}^{L} s_l^2 \right) \boldsymbol{I} \right), \tag{11}$$

and we will instead use the following prior distribution:

$$\mathcal{P}(\boldsymbol{Z}_{1:L}, \tilde{\boldsymbol{Z}}) = \prod_{i=1}^{d_z} \left( \prod_{l=1}^{L} P(\boldsymbol{z}_{l,i}) \right) p_{\boldsymbol{s}^2}(\tilde{\boldsymbol{z}}_i|\boldsymbol{Z}). \tag{12}$$

We now derive the ELBO using the newly established prior and posterior starting from

$$\log p_{\boldsymbol{\theta}}(\boldsymbol{x}) \geq -\mathcal{J}_{\text{RSQ-VAE}} = -\mathbb{E}_{\mathcal{Q}(\boldsymbol{Z}_{1:L}, \tilde{\boldsymbol{Z}}|\boldsymbol{x})} \left[ -\log p_{\boldsymbol{\theta}}(\boldsymbol{x}|\boldsymbol{Z}_{1:L}) + \log \frac{\mathcal{Q}(\boldsymbol{Z}_{1:L}, \tilde{\boldsymbol{Z}}|\boldsymbol{x})}{\mathcal{P}(\boldsymbol{Z}_{1:L}, \tilde{\boldsymbol{Z}})} \right]. \tag{13}$$

The above objective can be further simplified as

$$\mathcal{J}_{\text{RSQ-VAE}} = \frac{D}{2} \log \sigma^2 + \mathbb{E}_{\mathcal{Q}(\boldsymbol{Z}_{1:L}, \tilde{\boldsymbol{z}}_{1:L}|\boldsymbol{x})} \left[ \frac{\|\boldsymbol{x} - \boldsymbol{f}_{\boldsymbol{\theta}}(\boldsymbol{Z}_{1:L})\|_2^2}{2\sigma^2} + \frac{\|\tilde{\boldsymbol{Z}} - \boldsymbol{Z}\|_F^2}{2\sum_{l=1}^{L} s_l^2} - \sum_{l=1}^{L} H(\hat{P}_{s_l^2}(\boldsymbol{Z}_l|\tilde{\boldsymbol{Z}}_l)) \right], \tag{14}$$

where the third term is different from the one in Equation (10) and its numerator evaluates only the overall quantization error $\boldsymbol{H}_{\boldsymbol{\phi}}(\boldsymbol{x}) - \sum_{l=1}^{L} \boldsymbol{Z}_l$ with the dequantization process.

### 4.3.2 RSQ-VAE vs. RQ-VAE

RQ-VAE and RSQ-VAE both learn discrete representations in a coarse-to-fine manner, but RQ-VAE uses a deterministic RQ scheme to achieve the approximation in Equation (8), in which it is trained with the following objective function:

$$\mathcal{L}_{\text{RQ-VAE}} = \|\boldsymbol{x} - \boldsymbol{f}_{\boldsymbol{\theta}}(\boldsymbol{Z}_{1:L})\|_2^2 + \beta \sum_{l=1}^{L} \left\| \boldsymbol{H}_{\boldsymbol{\phi}}(\boldsymbol{x}) - \text{sg}\left[ \sum_{l'=1}^{l} \boldsymbol{Z}_{l'} \right] \right\|_F^2, \tag{15}$$

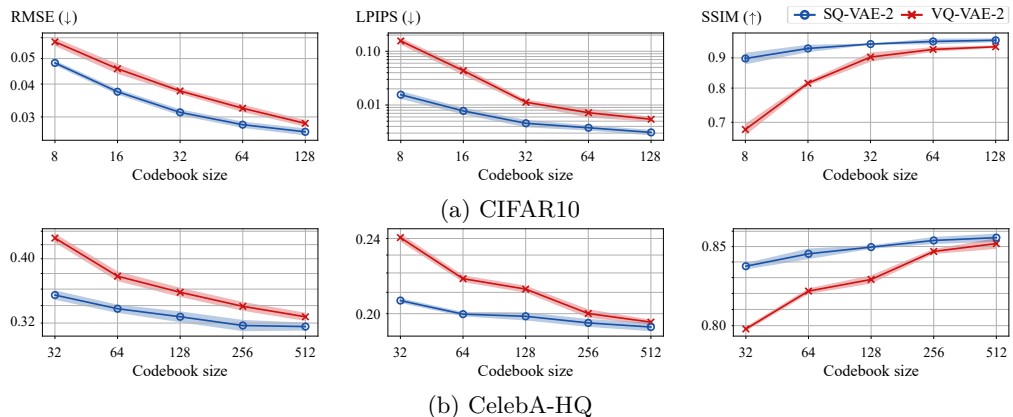

Figure 2: Impact of codebook capacity on reconstruction of images in (a) CIFAR10 and (b) CelebA-HQ. Two layers are tested on CIFAR10, three on CelebA-HQ.

Table 1: Evaluation on ImageNet ($256\times256$) and FFHQ ($1024\times1024$). RMSE ($\times10^2$), LPIPS, and SSIM are evaluated using the test set. Following Razavi et al. (2019), the codebook capacity for the discrete latent space is set to $(d_l, K_l) = (32^2, 512), (64^2, 512)$ and $(d_l, K_l) = (32^2, 512), (64^2, 512), (128^2, 512)$ for ImageNet and FFHQ, respectively. Codebook perplexity is also listed for each layer.

| Dataset | Model | Reconstruction | | | Codebook perplexity | | |
|---------|-------|--------|--------|--------|--------------------|--------------------|--------------------|
| | | RMSE↓ | LPIPS↓ | SSIM↑ | $\exp(H(Q(\boldsymbol{Z}_1)))$ | $\exp(H(Q(\boldsymbol{Z}_2)))$ | $\exp(H(Q(\boldsymbol{Z}_3)))$ |
| ImageNet | VQ-VAE-2 | $6.071 \pm 0.006$ | $0.265 \pm 0.012$ | $0.751 \pm 0.000$ | $106.8 \pm 0.8$ | $288.8 \pm 1.4$ | |
| | SQ-VAE-2 | $4.603 \pm 0.006$ | $0.096 \pm 0.000$ | $0.855 \pm 0.006$ | $406.2 \pm 0.9$ | $355.5 \pm 1.7$ | |
| FFHQ | VQ-VAE-2 | $4.866 \pm 0.291$ | $0.323 \pm 0.012$ | $0.814 \pm 0.003$ | $24.6 \pm 10.7$ | $41.3 \pm 14.0$ | $310.1 \pm 29.6$ |
| | SQ-VAE-2 | $2.118 \pm 0.013$ | $0.166 \pm 0.002$ | $0.909 \pm 0.001$ | $125.8 \pm 9.0$ | $398.7 \pm 14.1$ | $441.3 \pm 7.9$ |

where the codebooks are updated with the EMA update in the same manner as VQ-VAE. The second term of Equation (15) resembles the third term of Equation (10), which strongly enforces a certain degree of reconstruction even with only some of the information from the higher layers. RQ-VAE benefits from such a regularization term, which leads to stable training. However, in RSQ-VAE, this regularization degrades the reconstruction performance. To deal with this problem, we instead use Equation (14) as the objective, which regularizes the latent representation by taking into account of accumulated information from all layers.

**Remark.** HQ-VAE has favorable properties similar to those of SQ-VAE. The training scheme significantly reduces the number of hyperparameters required for training VQ-VAE variants. First, the objective functions (6) and (14) do not include any hyperparameters such as $\beta$ in Equations (7) and (15), except for the introduction of a temperature parameter for the Gumbel-softmax to enable the reparameterization of the discrete latent variables (Jang et al., 2017; Maddison et al., 2017). Please refer to Appendix E for the approximation of the categorical distributions $\mathcal{Q}(\boldsymbol{Z}_{1:L}, \tilde{\boldsymbol{Z}}_{1:L}|\boldsymbol{x})$. Moreover, our formulations do not require heuristic techniques such as stop-gradient, codebook reset, or EMA update. Additionally, HQ-VAE has several other empirical benefits over the VQ-VAE variants, such as the effect of self-annealing for better codebook utilization and a better rate-distortion (RD) trade-off, which will be verified in Section 5.

## 5 Experiments

We comprehensively examine SQ-VAE-2 and RSQ-VAE and their applicability to generative modeling. In particular, we compare SQ-VAE-2 with VQ-VAE-2 and RSQ-VAE with RQ-VAE to see if our framework improves reconstruction performance relative to the baselines. We basically compare various latent capacities in order to evaluate our methods in a RD sense (Alemi et al., 2017; Williams et al., 2020). The error plots the RD curves in the figures below indicate standard deviations based on four runs with different training seeds. In addition, we test HQ-VAE on an audio dataset to see if it is applicable to a different modality. Moreover, we investigate the characteristics of the *injected top-down* and *residual top-down* layers through

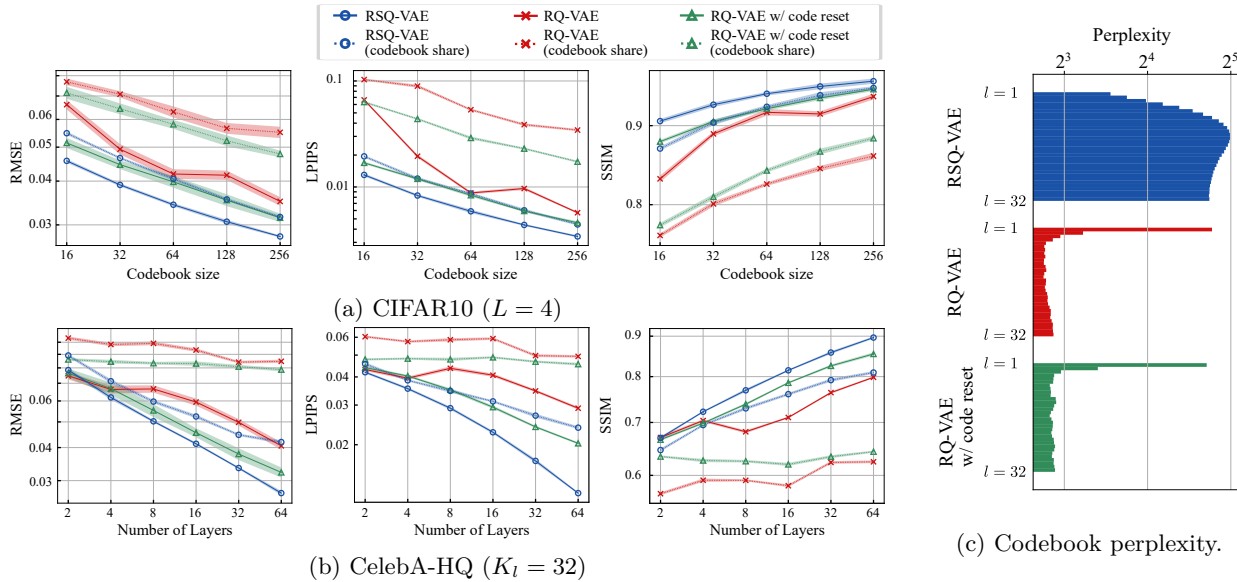

Figure 3: Impact of codebook capacity on reconstructions of images from (a) CIFAR10 and (b) CelebA-HQ. (c) Codebook perplexity at each layer is plotted, where models with 32 layers are trained on CelebA-HQ and all layers share the same codebook.

visualizations. Lastly, we apply HQ-VAEs to generative tasks to demonstrate their feasibility as feature extractors. Unless otherwise noted in what follows, we use the same network architecture in all models and set the codebook dimension to $d_b = 64$. Following the common dataset splits, we utilize the validation sets to adjust the learning rate during training and compute all the numerical metrics on the test sets. The details of these experiments are in Appendix F.

## 5.1 SQ-VAE-2 vs. VQ-VAE-2

We compare our SQ-VAE-2 with VQ-VAE-2 from the aspects of reconstruction accuracy and codebook utilization. First, we investigate their performance on CIFAR10 (Krizhevsky et al., 2009) and CelebA-HQ (256×256) under various codebook settings, i.e., different configurations for the hierarchical structure and numbers of code vectors ($K_l$). We evaluate the reconstruction accuracy in terms of a Euclidean metric and two perceptual metrics: the root mean squared error (RMSE), structure similarity index (SSIM) (Wang et al., 2004), and learned perceptual image patch similarity (LPIPS) (Zhang et al., 2018). As shown in Figure 2, SQ-VAE-2 achieves higher reconstruction accuracy in all cases. The difference in performance between the two models is especially noticeable when the codebook size is small.

**Comparison on large-scale datasets.** Next, we train SQ-VAE-2 and VQ-VAE-2 on ImageNet (256×256) (Deng et al., 2009) and FFHQ (1024×1024) (Karras et al., 2019) with the same latent settings as in Razavi et al. (2019). As shown in Table 1, SQ-VAE-2 achieves better reconstruction performance in terms of RMSE, LPIPS, and SSIM than VQ-VAE-2, which shows similar tendencies in the comparisons on CIFAR10 and CelebA-HQ. Furthermore, we measure codebook utilization per layer by using the perplexity of the latent variables. The codebook perplexity is defined as $\exp(H(Q(\boldsymbol{Z}_l)))$, where $Q(\boldsymbol{Z}_l)$ is a marginalized distribution of Equation (3) with $\boldsymbol{x} \sim p_d(\boldsymbol{x})$. The perplexity ranges from 1 to the number of code vectors ($K_l$) by definition. SQ-VAE-2 has higher codebook perplexities than VQ-VAE-2 at all layers. Here, VQ-VAE-2 did not effectively use the higher layers. In particular, the perplexity values at its top layer are extremely low, which is a sign of layer collapse.

## 5.2 RSQ-VAE vs. RQ-VAE

Table 2: Evaluation on UrbanSound8K. RMSE is evaluated on the test set. The network architecture follows the one described in Liu et al. (2021). Codebook size is set to $K_l = 8$.

| Model | Number of Layers | RMSE ↓ |
|---|---|---|
| RQ-VAE | 4 | $0.506 \pm 0.018$ |
| | 8 | $0.497 \pm 0.057$ |
| RSQ-VAE | 4 | $0.427 \pm 0.014$ |
| | 8 | $0.314 \pm 0.013$ |

(a) Reconstructed images and magnified differences of SQ-VAE-2

(b) $H(\hat{P}_{s_l^2}(\boldsymbol{z}_{l,i}|\tilde{\boldsymbol{Z}}_l))$ in SQ-VAE-2

(c) Reconstructed images and magnified differences of RSQ-VAE

(d) $H(\hat{P}_{s_l^2}(\boldsymbol{z}_{l,i}|\tilde{\boldsymbol{Z}}_l))$ in RSQ-VAE

Figure 4: Reconstructed samples with partial layers in (a) SQ-VAE-2 and (c) RSQ-VAE. The top row shows the reconstructed images, while the bottom row shows the components added at each layer. $(d_l, K_l) = (16^2, 256), (32^2, 16), (64^2, 4)$ and $(d_l, K_l) = (32^2, 4), (32^2, 16), (32^2, 256)$ for latent capacities, $l$ of 1, 2, and 3, respectively. Notice that the numbers of bits in these models are equal at each layer. For a reasonable visualization, we apply *progressive coding*, which induces progressive compression, to SQ-VAE-2 (see Appendix F.6). (b) and (d) Variance parameter $s_l^2$ normalized by the initial value $s_{l,0}^2$ and average entropy of the quantization process $(H(\hat{P}_{s_l^2}(\boldsymbol{z}_{l,i}|\tilde{\boldsymbol{Z}}_l)))$ at each layer.

We compare RSQ-VAE with RQ-VAE on the same metrics as in Section 5.1. As codebook reset is used in the original study of RQ-VAE (Zeghidour et al., 2021; Lee et al., 2022a) to prevent codebook collapse, we add a RQ-VAE incorporating this technique to the baselines. We do not apply it to RSQ-VAE because it is not explainable in the variational Bayes framework. In addition, Lee et al. (2022a) proposed that all the layers share the codebook, i.e., $\boldsymbol{B}_l = \boldsymbol{B}$ for $l \in [L]$, to enhance the utility of the codes. We thus test both RSQ-VAE and RQ-VAE with and without codebook sharing. First, we investigate their performances on CIFAR10 and CelebA-HQ ($256 \times 256$) by varying the number of quantization steps ($l$) and number of code vectors ($K_l$). As shown in Figures 3a and 3b, RSQ-VAE achieves higher reconstruction accuracy in terms of RMSE, SSIM, and LPIPS compared with the baselines, although the codebook reset overall improves the performance of RQ-VAE overall. The performance difference was remarkable when the codebook is shared by all layers. Moreover, there is a noticeable difference in how codes are used; more codes are assigned to the bottom layers in RSQ-VAE than to those in the RQ-VAEs (see Figure 3c). RSQ-VAE captures the coarse information with a relatively small number of codes and refines the reconstruction by allocating more bits at the bottom layers.

**Validation on an audio dataset.** We validate RSQ-VAE in the audio domain by comparing it with RQ-VAE in an experiment on reconstructing the normalized log-Mel spectrogram in an environmental sound dataset, UrbanSound8K (Salamon et al., 2014). We use the same network architecture as in an audio generation paper (Liu et al., 2021), in which multi-scale convolutional layers of varying kernel sizes are deployed to capture the local and global features of audio signals in the time-frequency domain (Xian et al.,

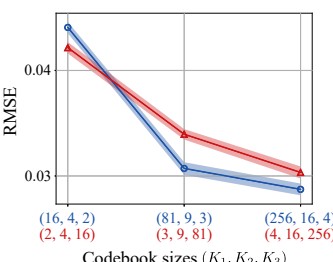 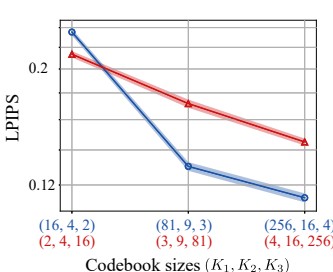 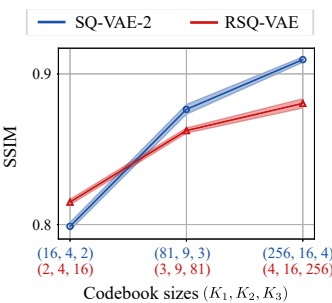

Figure 5: Comparison of SQ-VAE-2 and RSQ-VAE in three latent-capacity cases. The x-axis in blue (red) is $(K_1, K_2, K_3)$ for SQ-VAE-2 (RSQ-VAE). Note that $(d_1, d_2, d_3) = (16, 32, 64)$ for SQ-VAE-2 and $(d_1, d_2, d_3) = (32, 32, 32)$ for RSQ-VAE, and the values at the same point on the x-axis indicate the same latent capacity. SQ-VAE-2 outperforms RSQ-VAE at relatively large latent capacities. In contrast, RSQ-VAE achieves better reconstruction performance at higher compression rates.

2021). The codebook size is set to $K_l = 8$. The number of layers is set to 4 and 8, and all the layers share the same codebook. We run each trial with five different random seeds and obtain the average and standard deviation of the RMSEs. As shown in Table 2, RSQ-VAE has better average RMSEs than RQ-VAE has across different numbers of layers on the audio dataset. We also evaluate the results in terms of perceptual quality by performing subjective listening tests, which are described in Appendix F.2.3.

## 5.3 Empirical study of top-down layers

In this section, we focus on visualizing the obtained discrete representations. This will provide insights into the characteristics of the *top-down* layers. We train SQ-VAE-2 and RSQ-VAE, each with three layers, on CelebA-HQ (Karras et al., 2018). Figure 4 shows the progressively reconstructed images. For demonstration purposes, we incorporate *progressive coding* (Shu & Ermon, 2022) in SQ-VAE-2 to make the reconstructed images only with the top layers interpretable. Note that *progressive coding* is not applied to cases other than those illustrated in Figure 4. SQ-VAE-2 and RSQ-VAE share a similarity in that the higher layers generate the coarse part of the image while the lower layers complement them with details. However, upon examining Figure 4, we can see that, in the case of SQ-VAE-2, the additionally generated components (bottom row in Figure 4a) in each layer have different resolutions. We conjecture that the different layer-dependent resolutions $\boldsymbol{H}_{\boldsymbol{\phi}}^{r(l)}(\boldsymbol{x})$, which are injected into the *top-down* layers, contain different information. This implies that we may obtain more interpretable discrete representations if we can explicitly manipulate the extracted features in the *bottom-up* path to provide $\boldsymbol{H}_{\boldsymbol{\phi}}^{r(l)}(\boldsymbol{x})$, giving them more semantic meaning (e.g., texture or color). In contrast, RSQ-VAE seems to obtain a different discrete representation, which resembles more a decomposition. This might be due to its approximated expansion in Equation (8). Moreover, we can see from Figures 4b and 4d that the *top-down* layers also benefit from the *self-annealing* effect.

In Appendix F.3, we explore the idea of combining the two layers to form a hybrid model. There it is shown that the individual layers in a hybrid model produce effects similar to using them alone. That is, the outputs from the *injected top-down* layers have better resolution and *residual top-down* layers make refinements upon certain decompositions. Since these two layers enjoy distinct refining mechanisms, a hybrid model may bring a more flexible approximation to the posterior distribution.

Additionally, we compare the reconstruction performances of SQ-VAE-2 and RSQ-VAE with the same architecture for the *bottom-up* and *top-down* paths. The experimental conditions are the same as in the visualization of Figure 4. We examine three different compression rates by changing the number of code vectors $K_l$. Interestingly, Figure 5 shows that SQ-VAE-2 achieves better reconstruction performance in the case of the lower compression rate, whereas RSQ-VAE reconstructs the original images better than SQ-VAE-2 in the case of the higher compression rate.

Table 3: Image generation on FFHQ. † and ‡ denote the use of RQ-Transformer and contextual RQ-Transformer as a prior model.

| Model | FID↓ |
|---|---|
| VQ-GAN (Esser et al., 2021b) | 11.4 |
| RQ-VAE (Lee et al., 2022a) | 10.38 |
| RSQ-VAE† (ours) | 9.74 |
| RSQ-VAE‡ (ours) | 8.46 |

Table 4: Image generation on ImageNet.

| Model | FID↓ |
|---|---|
| VQ-VAE-2 (Razavi et al., 2019) | $\sim 31$ |
| DALL-E (Ramesh et al., 2021) | 32.01 |
| VQ-GAN (Esser et al., 2021b) | 15.78 |
| RQ-VAE (Lee et al., 2022a) | 7.55 |
| HQ-TVAE (You et al., 2022) | 7.15 |
| Contextual RQ-Transformer (Lee et al., 2022b) | 3.41 |
| SQ-VAE-2 (ours) | 4.51 |

### 5.4 Applications of HQ-VAEs to generative tasks

Lastly, we conduct experiments in the vision domain to demonstrate the applicability of HQ-VAE to realistic generative tasks.

First, we train RSQ-VAE on FFHQ ($256 \times 256$) by using the same encoder–decoder architecture as in Lee et al. (2022a). We set the latent capacities to $L = 4$, $(d_l, K_l) = (8^2, 256)$ for $l \in [4]$ by following their RQ-VAE. We train prior models, an RQ-Transformer (Lee et al., 2022a) and a contextual RQ-Transformer (Lee et al., 2022b), on the latent feature extracted by RSQ-VAE. Table 3 lists the Fréchet inception distance (FID) scores for the generated images (refer to Table 7 for more detailed comparisons). According to the table, our RSQ-VAE leads to comparable generation performance to RQ-VAE, even without adversarial training. Figure 16 shows the generated samples from our model on FFHQ.

Next, we train SQ-VAE-2 on ImageNet ($256 \times 256$) by using a simple resblock-based encoder–decoder architecture. We define two levels of discrete latent spaces whose capacities are $(d_1, K_1) = (16^2, 2048)$ and $(d_2, K_2) = (32^2, 1024)$. For training SQ-VAE-2 training, we use the adversarial training framework to achieve a perceptual reconstruction with a feasible compression ratio. Subsequently, we use Muse (Chang et al., 2023) to train the prior model on the hierarchical discrete latent features. As reported in Table 4, our SQ-VAE-2-based models achieve FID score that is competitive with current state-of-the-art models (refer to Table 8 for more detailed comparisons). Figure 17 shows the generated images from our model on ImageNet.

## 6 Discussion

### 6.1 Conclusion

We proposed HQ-VAE, a general VAE approach that learns hierarchical discrete representations. HQ-VAE is formulated within the variational Bayes framework as a stochastic quantization technique, which (1) greatly reduces the number of hyperparameters to be tuned, and (2) enhances codebook usage without any heuristics thanks to the *self-annealing* effect. We instantiated the general HQ-VAE with two types of posterior approximators for the discrete latent representations (SQ-VAE-2 and RSQ-VAE). These two variants have infomartion passing designs similar to those of as VQ-VAE-2 and RQ-VAE, respectively, but their latent representations are quantized stochastically. Our experiments show that SQ-VAE-2 and RSQ-VAE outperformed their individual baselines with better reconstruction and more efficient codebook usages in the image domain as well as the audio domain.

### 6.2 Concluding remarks

Replacing VQ-VAE-2 and RQ-VAE with SQ-VAE-2 and RSQ-VAE will yeild comparative improvements to many of the previous methods in terms of reconstruction accuracy and efficient codebook usage. SQ-VAE-2 and RSQ-VAE have better RD curves than the baselines have, which means that they achieve (i) better reconstruction performance for the same latent capacities, and (ii) comparable reconstruction performance at higher compression rate. Furthermore, our approach eliminates the need for repetited tunings of many hyper-parameters or the use of ad-hoc techniques. SQ-VAE-2 and RSQ-VAE are applicable to generative modeling with additional training of prior models as was done in numerous studies. Generally, compression

models with better RD curves are more feasible for the prior models (Rombach et al., 2022); hence, the replacement of VQ-VAE-2 (RQ-VAE) with SQ-VAE-2 (RSQ-VAE) would be beneficial even for generative modeling. Recently, RQ-VAE has been used more often in generation tasks than VQ-VAE-, wihch has a severe instability issue, i.e., layer collapse (Dhariwal et al., 2020). Here, we believe that SQ-VAE-2 has potential as a hierarchical model for generation tasks since it mitigates the issue greatly.

SQ-VAE-2 and RSQ-VAE have their own unique advantages as follows. SQ-VAE-2 can learn multi-resolution discrete representations, thanks to the design of the *bottom-up* path with the pooling operators (see Figure 4a). The results of this study imply that further semantic disentanglement of the discrete representation might be possible by including specific inductive architectural components in the *bottom-up* path. Moreover, SQ-VAE-2 outperforms RSQ-VAE at lower compression rates (see Section 5.3). This hints at that SQ-VAE-2 might be appropriate especially for high-fidelity generation tasks when the prior model is large enough. In contrast, one of the strengths of RSQ-VAE is that it can easily accommodate different compression rates simply by changing the number of layers during the inference (without changing or retraining the model). Furthermore, RSQ-VAE outperforms SQ-VAE-2 at higher compression rates (see Section 5.3). These properties would make RSQ-VAE suitable for application to neural codecs (Zeghidour et al., 2021; Défossez et al., 2022).

## Acknowledgement

We would like to thank Marc Ferras for many helpful comments during the discussion phase. Besides, we thank anonymous reviewers for their valuable suggestions and comments.

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

## Contents

## A Literature of stochastic posterior modeling for discrete latent

We briefly compare SQ-VAE, the foundation of HQ-VAE, with two other stochastic discrete models.

Sønderby et al. (2017) proposed stochastic posterior modeling in the discrete latent setting as $Q(\boldsymbol{z}_i = \boldsymbol{b}_k|\boldsymbol{x}) \propto -\|\tilde{\boldsymbol{z}}_i - \boldsymbol{b}_k\|_2^2$, followed by succeeding works (Roy et al., 2018; Williams et al., 2020). Although the approach resembles SQ-VAE in terms of stochastic quantization, they have non-trivial differences as follows. In SQ-VAE, Takida et al. (2022b) introduced a trainable parameter, denoted as $\boldsymbol{\Sigma}_{\varphi}$, by considering a dequantization process as the inverse operation of quantization. This leads to $Q(\boldsymbol{z}_i = \boldsymbol{b}_k|\boldsymbol{x}) \propto -\frac{1}{2}(\tilde{\boldsymbol{z}}_i - \boldsymbol{b}_k)^{\top}\boldsymbol{\Sigma}_{\varphi}^{-1}(\tilde{\boldsymbol{z}}_i - \boldsymbol{b}_k)$, which imposes the self-annealing effect as well as allows the posterior modeling with more general distances. Please refer to Takida et al. (2022b) for more details about the difference and some numerical examples.

The logits modeling (LM) in dVAE (Ramesh et al., 2021) is another approach to model the discrete posterior distribution $(Q(\mathbf{Z}|\mathbf{x}))$ instead of vector quantization. LM assumes an encoder that directly maps from $\mathbf{x}$ to the logits of $Q(\mathbf{z}_i|\mathbf{x})$. Therefore, it does not have raw explicit features $\tilde{\mathbf{Z}}$. On the other hand, in SQ-VAE, the encoded features $\tilde{\mathbf{Z}}$ are directly quantized. When the learnable variance parameter $s^2$ is set (fixed) close to 0, conventional VQ is obtained. We used SQ instead of LM for the posterior modeling in HQ-VAE because SQ can serve as a drop-in replacement of VQ and can be extended to hierarchical and residual quantizations similar to VQ.

## B From SQ-VAE to HQ-VAE

Let $\boldsymbol{B} \in \{\boldsymbol{b}_k\}_{k=1}^K$ $(\boldsymbol{b}_k \in \mathbb{R}^{d_b})$ be a trainable codebook and $\boldsymbol{Z} \in \boldsymbol{B}^d$ be a discrete latent variable to represent the target data $\boldsymbol{x}$. An auxiliary variable, denoted as $\tilde{\boldsymbol{Z}} \in \mathbb{R}^{d_b \times d}$, is used to connect $\boldsymbol{x}$ and $\boldsymbol{Z}$ in the SQ framework. In a general VAE with discrete and auxiliary latent variables, the generative process is modeled as $\boldsymbol{x} \sim p_{\boldsymbol{\theta}}(\boldsymbol{x}|\boldsymbol{Z})$ with a prior distribution $\boldsymbol{Z}, \tilde{\boldsymbol{Z}}) \sim \mathcal{P}(\boldsymbol{Z}, \tilde{\boldsymbol{Z}})$. For variational inference, a variational distribution,

denoted as $\mathcal{Q}(\boldsymbol{Z}, \tilde{\boldsymbol{Z}}|\boldsymbol{x})$, is used to approximate the posterior distribution $\mathcal{P}(\boldsymbol{Z}, \tilde{\boldsymbol{Z}}|\boldsymbol{x})$. The ELBO becomes

$$
\begin{aligned}
\log p_{\boldsymbol{\theta}}(\boldsymbol{x}) &\geq \log p_{\boldsymbol{\theta}}(\boldsymbol{x}) - D_{\mathrm{KL}}(\mathcal{Q}(\boldsymbol{Z}, \tilde{\boldsymbol{Z}}|\boldsymbol{x}) \parallel \mathcal{P}(\boldsymbol{Z}, \tilde{\boldsymbol{Z}}|\boldsymbol{x})) \\
&= \mathbb{E}_{\mathcal{Q}(\boldsymbol{Z}, \tilde{\boldsymbol{Z}}|\boldsymbol{x})} \left[ \log \frac{p_{\boldsymbol{\theta}}(\boldsymbol{x})\mathcal{P}(\boldsymbol{Z}, \tilde{\boldsymbol{Z}}|\boldsymbol{x})}{\mathcal{Q}(\boldsymbol{Z}, \tilde{\boldsymbol{Z}}|\boldsymbol{x})} \right] \\
&= \mathbb{E}_{\mathcal{Q}(\boldsymbol{Z}, \tilde{\boldsymbol{Z}}|\boldsymbol{x})} \left[ \log p_{\boldsymbol{\theta}}(\boldsymbol{x}|\boldsymbol{Z}) - \log \frac{\mathcal{Q}(\boldsymbol{Z}, \tilde{\boldsymbol{Z}}|\boldsymbol{x})}{\mathcal{P}(\boldsymbol{Z}, \tilde{\boldsymbol{Z}})} \right].
\end{aligned}
$$

In the vanilla SQ-VAE (Takida et al., 2022b), the vectors composing $\boldsymbol{Z}$ are assumed to be independent of each other and the quantization process is applied to each vector such that $\mathcal{P}(\boldsymbol{Z}, \tilde{\boldsymbol{Z}}) = \prod_{i=1}^{d} P(\boldsymbol{z}_i)p(\tilde{\boldsymbol{z}}_i|\boldsymbol{z}_i)$ and $\mathcal{Q}(\boldsymbol{Z}, \tilde{\boldsymbol{Z}}|\boldsymbol{x}) = \prod_{i=1}^{d} q(\tilde{\boldsymbol{z}}_i|\boldsymbol{x})\hat{P}(\boldsymbol{z}_i|\tilde{\boldsymbol{z}}_i)$. In contrast, HQ-VAE allows dependencies in $\boldsymbol{Z}$ to be modelled by decomposing it into $L$ groups, i.e., $\boldsymbol{Z} = \{\boldsymbol{Z}_l\}_{l=1}^{L}$ and $\boldsymbol{Z}_l \in \boldsymbol{B}^{d_l}$ ($d = \sum_{l=1}^{L} d_l$), and modeling dependencies between $\{\boldsymbol{Z}_l\}_{l=1}^{L}$ by designing the approximated posterior $\mathcal{Q}(\boldsymbol{Z}, \tilde{\boldsymbol{Z}}|\boldsymbol{x})$ in decomposed ways. This paper provides instances in the form of *top-down* blocks to model the dependencies, which enables graphical models to constructed for $(\boldsymbol{x}, \{\boldsymbol{Z}_l\}_{l=1}^{L})$ in a handy way (see Figure 1a). This framework naturally covers the latent structures in VQ-VAE-2 and RQ-VAE (see Sections 4.2.2 and 4.3.2).

## C  Derivations

### C.1  SQ-VAE-2

The ELBO of SQ-VAE-2 is formulated by using Bayes' theorem:

$$
\begin{aligned}
\log p_{\boldsymbol{\theta}}(\boldsymbol{x}) &\geq \log p_{\boldsymbol{\theta}}(\boldsymbol{x}) - D_{\mathrm{KL}}(\mathcal{Q}(\boldsymbol{Z}_{1:L}, \tilde{\boldsymbol{Z}}_{1:L}|\boldsymbol{x}) \parallel \mathcal{P}(\boldsymbol{Z}_{1:L}, \tilde{\boldsymbol{Z}}_{1:L}|\boldsymbol{x})) \\
&= \mathbb{E}_{\mathcal{Q}(\boldsymbol{Z}_{1:L}, \tilde{\boldsymbol{Z}}_{1:L}|\boldsymbol{x})} \left[ \log \frac{p_{\boldsymbol{\theta}}(\boldsymbol{x})\mathcal{P}(\boldsymbol{Z}_{1:L}, \tilde{\boldsymbol{Z}}_{1:L}|\boldsymbol{x})}{\mathcal{Q}(\boldsymbol{Z}_{1:L}, \tilde{\boldsymbol{Z}}_{1:L}|\boldsymbol{x})} \right] \\
&= \mathbb{E}_{\mathcal{Q}(\boldsymbol{Z}_{1:L}, \tilde{\boldsymbol{Z}}_{1:L}|\boldsymbol{x})} \left[ \log p_{\boldsymbol{\theta}}(\boldsymbol{x}|\boldsymbol{Z}_{1:L}) - \log \frac{\mathcal{Q}(\boldsymbol{Z}_{1:L}, \tilde{\boldsymbol{Z}}_{1:L}|\boldsymbol{x})}{\mathcal{P}(\boldsymbol{Z}_{1:L}, \tilde{\boldsymbol{Z}}_{1:L})} \right] \\
&= \mathbb{E}_{\mathcal{Q}(\boldsymbol{Z}_{1:L}, \tilde{\boldsymbol{Z}}_{1:L}|\boldsymbol{x})} \left[ \log p_{\boldsymbol{\theta}}(\boldsymbol{x}|\boldsymbol{Z}_{1:L}) - \sum_{l=1}^{L}\sum_{i=1}^{d_l} \left( \log \frac{p_{s_l^2}(\tilde{\boldsymbol{z}}_{l,i}|\hat{\boldsymbol{Z}}_l)}{p_{s_l^2}(\tilde{\boldsymbol{z}}_{l,i}|\boldsymbol{Z}_l)} + \log \frac{\hat{P}_{s_l^2}(\boldsymbol{z}_{l,i}|\tilde{\boldsymbol{Z}}_l)}{P(\boldsymbol{z}_{l,i})} \right) \right] \\
&= \mathbb{E}_{\mathcal{Q}(\boldsymbol{Z}_{1:L}, \tilde{\boldsymbol{Z}}_{1:L}|\boldsymbol{x})} \left[ \log p_{\boldsymbol{\theta}}(\boldsymbol{x}|\boldsymbol{Z}_{1:L}) + \sum_{l=1}^{L}\sum_{i=1}^{d_l} \left( \log \frac{p_{s_l^2}(\tilde{\boldsymbol{z}}_{l,i}|\boldsymbol{Z}_l)}{p_{s_l^2}(\tilde{\boldsymbol{z}}_{l,i}|\hat{\boldsymbol{Z}}_l)} + H(\hat{P}_{s_l^2}(\boldsymbol{z}_{l,i}|\tilde{\boldsymbol{Z}}_l)) - \log K_l \right) \right]. \quad (16)
\end{aligned}
$$

Since the probabilistic parts are modeled as Gaussian distributions, the first and second terms can be calculated as

$$
\begin{aligned}
\log p_{\boldsymbol{\theta}}(\boldsymbol{x}|\boldsymbol{Z}_{1:L}) &= \log \mathcal{N}(\boldsymbol{x}; \boldsymbol{f}_{\boldsymbol{\theta}}(\boldsymbol{Z}_{1:L}), \sigma^2 \boldsymbol{I}) \\
&= -\frac{D}{2}\log(2\pi\sigma^2) - \frac{1}{2\sigma^2}\|\boldsymbol{x} - \boldsymbol{f}_{\boldsymbol{\theta}}(\boldsymbol{x})\|_2^2 \quad \text{and}
\end{aligned} \quad (17)
$$

$$
\begin{aligned}
\mathbb{E}_{\mathcal{Q}(\boldsymbol{Z}_{1:L}, \tilde{\boldsymbol{Z}}_{1:L}|\boldsymbol{x})} \left[ \frac{p_{s_l^2}(\tilde{\boldsymbol{z}}_{l,i}|\boldsymbol{Z}_l)}{p_{s_l^2}(\tilde{\boldsymbol{z}}_{l,i}|\hat{\boldsymbol{Z}}_l)} \right] &= \mathbb{E}_{\mathcal{Q}(\boldsymbol{Z}_{1:L}, \tilde{\boldsymbol{Z}}_{1:L}|\boldsymbol{x})} \left[ -\frac{1}{2s_l^2}\|\tilde{\boldsymbol{z}}_{l,i} - \boldsymbol{z}_{l,i}\|_2^2 + \frac{1}{2s_l^2}\|\tilde{\boldsymbol{z}}_{l,i} - \hat{\boldsymbol{z}}_{l,i}\|_2^2 \right] \\
&= -\mathbb{E}_{\mathcal{Q}(\boldsymbol{Z}_{1:L}, \tilde{\boldsymbol{Z}}_{1:L}|\boldsymbol{x})} \left[ \frac{1}{2s_l^2}\|\tilde{\boldsymbol{z}}_{l,i} - \boldsymbol{z}_{l,i}\|_2^2 \right] + \frac{d_b}{2}. \quad (18)
\end{aligned}
$$

By substituting Equations (17) and (18) into Equation (16), we arrive at Equation (6), where we use $\tilde{\boldsymbol{Z}}_l = \hat{\boldsymbol{Z}}_l$ instead of sampling it in a practical implementation.

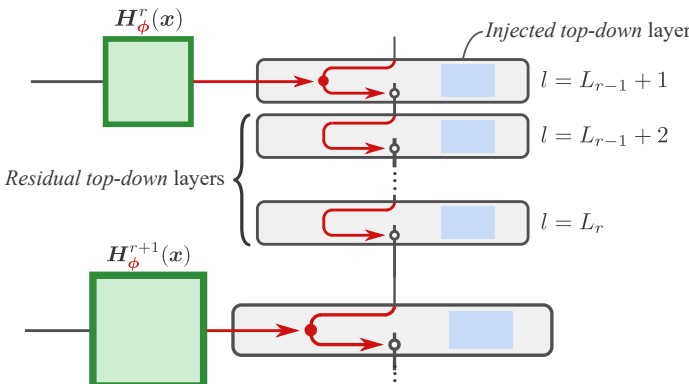

Figure 6: *Top-down* layers corresponding to the $r$th resolution of the hybrid model in Appendix D.

## C.2 RSQ-VAE

The ELBO of RSQ-VAE is formulated by using Bayes' theorem:

$$\log p_{\boldsymbol{\theta}}(\boldsymbol{x}) \geq \log p_{\boldsymbol{\theta}}(\boldsymbol{x}) - D_{\mathrm{KL}}(\mathcal{Q}(\boldsymbol{Z}_{1:L}, \tilde{\boldsymbol{Z}}|\boldsymbol{x}) \parallel \mathcal{P}(\boldsymbol{Z}_{1:L}, \tilde{\boldsymbol{Z}}|\boldsymbol{x}))$$

$$= \mathbb{E}_{\mathcal{Q}(\boldsymbol{Z}_{1:L}, \tilde{\boldsymbol{z}}_{1:L}|\boldsymbol{x})} \left[ \log \frac{p_{\boldsymbol{\theta}}(\boldsymbol{x})\mathcal{P}(\boldsymbol{Z}_{1:L}, \tilde{\boldsymbol{Z}}|\boldsymbol{x})}{\mathcal{Q}(\boldsymbol{Z}_{1:L}, \tilde{\boldsymbol{Z}}|\boldsymbol{x})} \right]$$

$$= \mathbb{E}_{\mathcal{Q}(\boldsymbol{Z}_{1:L}, \tilde{\boldsymbol{Z}}|\boldsymbol{x})} \left[ \log p_{\boldsymbol{\theta}}(\boldsymbol{x}|\boldsymbol{Z}_{1:L}) - \log \frac{\mathcal{Q}(\boldsymbol{Z}_{1:L}, \tilde{\boldsymbol{Z}}|\boldsymbol{x})}{\mathcal{P}(\boldsymbol{Z}_{1:L}, \tilde{\boldsymbol{Z}})} \right]$$

$$= \mathbb{E}_{\mathcal{Q}(\boldsymbol{Z}_{1:L}, \tilde{\boldsymbol{Z}}|\boldsymbol{x})} \left[ \log p_{\boldsymbol{\theta}}(\boldsymbol{x}|\boldsymbol{Z}_{1:L}) - \sum_{i=1}^{d_l} \log \frac{p_{\boldsymbol{s}^2}(\tilde{\boldsymbol{z}}_i|\hat{\boldsymbol{Z}})}{p_{\boldsymbol{s}^2}(\tilde{\boldsymbol{z}}_i|\boldsymbol{Z})} - \sum_{l=1}^{L}\sum_{i=1}^{d_l} \log \frac{\hat{P}_{s_l^2}(\boldsymbol{z}_{l,i}|\tilde{\boldsymbol{Z}}_l)}{P(\boldsymbol{z}_{l,i})} \right]$$

$$= \mathbb{E}_{\mathcal{Q}(\boldsymbol{Z}_{1:L}, \tilde{\boldsymbol{Z}}|\boldsymbol{x})} \left[ \log p_{\boldsymbol{\theta}}(\boldsymbol{x}|\boldsymbol{Z}_{1:L}) + \sum_{i=1}^{d_l} \log \frac{p_{\boldsymbol{s}^2}(\tilde{\boldsymbol{z}}_i|\boldsymbol{Z})}{p_{\boldsymbol{s}^2}(\tilde{\boldsymbol{z}}_i|\hat{\boldsymbol{Z}})} + \sum_{l=1}^{L}\sum_{i=1}^{d_l} H(\hat{P}_{s_l^2}(\boldsymbol{z}_{l,i}|\tilde{\boldsymbol{Z}}_l)) - \log K_l \right]. \quad (19)$$

Since the probabilistic parts are modeled as Gaussian distributions, the second term can be calculated as

$$\mathbb{E}_{\mathcal{Q}(\boldsymbol{Z}_{1:L}, \tilde{\boldsymbol{Z}}|\boldsymbol{x})} \left[ \frac{p_{\boldsymbol{s}^2}(\tilde{\boldsymbol{z}}_i|\boldsymbol{Z})}{p_{\boldsymbol{s}^2}(\tilde{\boldsymbol{z}}_i|\hat{\boldsymbol{Z}})} \right] = \mathbb{E}_{\mathcal{Q}(\boldsymbol{Z}_{1:L}, \tilde{\boldsymbol{z}}|\boldsymbol{x})} \left[ -\frac{1}{2\sum_{l=1}^{L} s_l^2} \|\tilde{\boldsymbol{z}}_i - \boldsymbol{z}_i\|_2^2 + \frac{1}{2\sum_{l=1}^{L} s_l^2} \|\tilde{\boldsymbol{z}}_i - \hat{\boldsymbol{z}}_i\|_2^2 \right]$$

$$= -\mathbb{E}_{\mathcal{Q}(\boldsymbol{Z}_{1:L}, \tilde{\boldsymbol{z}}|\boldsymbol{x})} \left[ \frac{1}{2\sum_{l=1}^{L} s_l^2} \|\tilde{\boldsymbol{z}}_i - \boldsymbol{z}_i\|_2^2 \right] + \frac{d_b}{2}. \quad (20)$$

By substituting Equations (17) and (20) into (19), we find that Equation (14), where we use $\tilde{\boldsymbol{Z}} = \boldsymbol{H}_{\boldsymbol{\phi}}(\boldsymbol{x})$ instead of sampling it in a practical implementation. The above is the derivation of the ELBO objective in the case of $R = 1$. Appendix D extends this model to the general case with a variable $R$, which is equivalent to the hybrid model.

## D Hybrid model

Here, we describe the ELBO of a hybrid model where the two types of *top-down* layers are combinatorially used to build a *top-down* path as in Figure 6. Some extra notation will be needed as follows: $L_r$ indicates the number of layers corresponding to the resolutions from the first to $r$th order; $\ell_r := \{L_{r-1}+1, \cdots, L_r\}$ is the set of layers corresponding to the resolution $r$; and the output of the encoding block in the $(L_{r-1}+1)$th layer is denoted as $\tilde{\boldsymbol{G}}_{\boldsymbol{\phi}}^r(\boldsymbol{H}_{\boldsymbol{\phi}}^r(\boldsymbol{x}), \boldsymbol{Z}_{1:L_{r-1}})$. In Figure 6, the quantized variables $\boldsymbol{Z}_{\ell_r}$ aim at approximating the variable encoded at $l = L_{r-1}+1$ as

$$\tilde{\boldsymbol{G}}_{\boldsymbol{\phi}}^r(\boldsymbol{H}_{\boldsymbol{\phi}}^r(\boldsymbol{x}), \boldsymbol{Z}_{1:L_{r-1}}) \approx \sum_{l \in \ell_r} \boldsymbol{Z}_l =: \boldsymbol{Y}_r. \quad (21)$$

On this basis, the $l$th *top-down* layer quantizes the following information:

$$\hat{\boldsymbol{Z}}_l = \boldsymbol{G}_{\boldsymbol{\phi}}^l(\boldsymbol{x}, \boldsymbol{Z}_{1:l-1}) = \begin{cases} \boldsymbol{H}_{\boldsymbol{\phi}}^1(\boldsymbol{x}) & (l = 1) \\ \tilde{\boldsymbol{G}}_{\boldsymbol{\phi}}^{r(l)}(\boldsymbol{H}_{\boldsymbol{\phi}}^{r(l)}(\boldsymbol{x}), \boldsymbol{Z}_{1:L_{r(l)-1}}) - \sum_{l'=L_{r(l)-1}+1}^l \boldsymbol{Z}_{l'} & (l > 1). \end{cases} \tag{22}$$

To derive the ELBO objective, we consider conditional distributions on $(\boldsymbol{Z}_{1:L}, \tilde{\boldsymbol{Y}}_{1:R})$, where $\tilde{\boldsymbol{Y}}_r := \sum_{l \in \ell_r} \tilde{\boldsymbol{Z}}_l$. From the reproductive property of the Gaussian distribution, the continuous latent variable converted from $\boldsymbol{Y}_r$ via the stochastic dequantization processes, $\tilde{\boldsymbol{Z}}_{\ell_r}$, follows a Gaussian distribution:

$$p_{\boldsymbol{s}_r^2}(\tilde{\boldsymbol{y}}_{r,i}|\boldsymbol{Z}_{\ell_r}) = \mathcal{N}\left(\tilde{\boldsymbol{y}}_{r,i}; \sum_{l \in \ell_r} \boldsymbol{z}_{l,i}, \left(\sum_{l \in \ell_r} s_l^2\right)\boldsymbol{I}\right), \tag{23}$$

where $\boldsymbol{s}_r^2 := \{s_l^2\}_{l \in \ell_r}$. We will use the following prior distribution to derive the ELBO objective:

$$\mathcal{P}(\boldsymbol{Z}_{1:L}, \tilde{\boldsymbol{Y}}_{1:R}) = \prod_{r=1}^R \prod_{i=1}^{d_r} \left(\prod_{l \in \ell_r} P(\boldsymbol{z}_{l,i})\right) p_{\boldsymbol{s}_r^2}(\tilde{\boldsymbol{y}}_{r,i}|\boldsymbol{Z}_{\ell_r}), \tag{24}$$

where $d_r := d_l$ for $l \in \ell_r$. With the prior and posterior distributions, the ELBO of the hybrid model can be formulated by invoking Bayes' theorem:

$$
\begin{aligned}
\log p_{\boldsymbol{\theta}}(\boldsymbol{x}) &\geq \log p_{\boldsymbol{\theta}}(\boldsymbol{x}) - D_{\mathrm{KL}}(\mathcal{Q}(\boldsymbol{Z}_{1:L}, \tilde{\boldsymbol{Y}}_{1:R}|\boldsymbol{x}) \,\|\, \mathcal{P}(\boldsymbol{Z}_{1:L}, \tilde{\boldsymbol{Y}}_{1:R}|\boldsymbol{x})) \\
&= \mathbb{E}_{\mathcal{Q}(\boldsymbol{Z}_{1:L}, \tilde{\boldsymbol{Y}}_{1:R}|\boldsymbol{x})}\left[\log \frac{p_{\boldsymbol{\theta}}(\boldsymbol{x})\mathcal{P}(\boldsymbol{Z}_{1:L}, \tilde{\boldsymbol{Y}}_{1:R}|\boldsymbol{x})}{\mathcal{Q}(\boldsymbol{Z}_{1:L}, \tilde{\boldsymbol{Y}}_{1:R}|\boldsymbol{x})}\right] \\
&= \mathbb{E}_{\mathcal{Q}(\boldsymbol{Z}_{1:L}, \tilde{\boldsymbol{Y}}_{1:R}|\boldsymbol{x})}\left[\log p_{\boldsymbol{\theta}}(\boldsymbol{x}|\boldsymbol{Z}_{1:L}) - \log \frac{\mathcal{Q}(\boldsymbol{Z}_{1:L}, \tilde{\boldsymbol{Y}}_{1:R}|\boldsymbol{x})}{\mathcal{P}(\boldsymbol{Z}_{1:L}, \tilde{\boldsymbol{Y}}_{1:R})}\right] \\
&= \mathbb{E}_{\mathcal{Q}(\boldsymbol{Z}_{1:L}, \tilde{\boldsymbol{Y}}_{1:R}|\boldsymbol{x})}\left[\log p_{\boldsymbol{\theta}}(\boldsymbol{x}|\boldsymbol{Z}_{1:L}) - \sum_{r=1}^R \sum_{i=1}^{d_r} \log \frac{p_{\boldsymbol{s}_r^2}(\tilde{\boldsymbol{y}}_{r,i}|\hat{\boldsymbol{Z}}_{\ell_r})}{p_{\boldsymbol{s}_r^2}(\tilde{\boldsymbol{y}}_{r,i}|\boldsymbol{Z}_{\ell_r})} - \sum_{l=1}^L \sum_{i=1}^{d_l} \log \frac{\hat{P}_{s_l^2}(\boldsymbol{z}_{l,i}|\tilde{\boldsymbol{Z}}_l)}{P(\boldsymbol{z}_{l,i})}\right] \\
&= \mathbb{E}_{\mathcal{Q}(\boldsymbol{Z}_{1:L}, \tilde{\boldsymbol{Y}}_{1:R}|\boldsymbol{x})}\left[\log p_{\boldsymbol{\theta}}(\boldsymbol{x}|\boldsymbol{Z}_{1:L}) + \sum_{r=1}^R \sum_{i=1}^{d_r} \log \frac{p_{\boldsymbol{s}_r^2}(\tilde{\boldsymbol{y}}_{r,i}|\boldsymbol{Z}_{\ell_r})}{p_{\boldsymbol{s}_r^2}(\tilde{\boldsymbol{y}}_{r,i}|\hat{\boldsymbol{Z}}_{\ell_r})} + \sum_{l=1}^L \sum_{i=1}^{d_l} H(\hat{P}_{s_l^2}(\boldsymbol{z}_{l,i}|\tilde{\boldsymbol{Z}}_l)) - \log K_l\right],
\end{aligned}
\tag{25}
$$

where $\hat{\boldsymbol{Y}}_r = \tilde{\boldsymbol{G}}_{\boldsymbol{\phi}}^r(\boldsymbol{H}_{\boldsymbol{\phi}}^r(\boldsymbol{x}), \boldsymbol{Z}_{1:L_{r-1}})$. Since we have modelled the dequantization process and the probabilistic decoder as Gaussians, by substituting their closed forms into the above equation, we find that

$$
\begin{aligned}
\mathcal{J}_{\text{HQ-VAE}} = {}& \frac{D}{2}\log\sigma^2 \\
&+ \mathbb{E}_{\mathcal{Q}(\boldsymbol{Z}_{1:L}, \tilde{\boldsymbol{Z}}_{1:L}|\boldsymbol{x})}\left[\frac{\|\boldsymbol{x} - \boldsymbol{f}_{\boldsymbol{\theta}}(\boldsymbol{Z}_{1:L})\|_2^2}{2\sigma^2} + \sum_{r=1}^R \frac{\left\|\tilde{\boldsymbol{G}}_{\boldsymbol{\phi}}^r(\boldsymbol{H}_{\boldsymbol{\phi}}^r(\boldsymbol{x}), \boldsymbol{Z}_{1:L_{r-1}}) - \sum_{l \in \ell_r} \boldsymbol{Z}_l\right\|_F^2}{2\sum_{l \in \ell_r} s_l^2} - \sum_{l=1}^L H(\hat{P}_{s_l^2}(\boldsymbol{Z}_l|\tilde{\boldsymbol{Z}}_l))\right],
\end{aligned}
\tag{26}
$$

where we have used

$$
\begin{aligned}
\mathbb{E}_{\mathcal{Q}(\boldsymbol{Z}_{1:L}, \tilde{\boldsymbol{Y}}_{1:R}|\boldsymbol{x})}\left[\frac{p_{\boldsymbol{s}_r^2}(\tilde{\boldsymbol{y}}_{r,i}|\boldsymbol{Z}_{\ell_r})}{p_{\boldsymbol{s}_r^2}(\tilde{\boldsymbol{y}}_{r,i}|\hat{\boldsymbol{Z}}_{\ell_r})}\right] &= \mathbb{E}_{\mathcal{Q}(\boldsymbol{Z}_{1:L}, \tilde{\boldsymbol{Y}}_{1:R}|\boldsymbol{x})}\left[-\frac{1}{2\sum_{l \in \ell_r} s_l^2}\|\tilde{\boldsymbol{y}}_{r,i} - \boldsymbol{y}_{r,i}\|_2^2 + \frac{1}{2\sum_{l \in \ell_r} s_l^2}\|\tilde{\boldsymbol{y}}_{r,i} - \hat{\boldsymbol{y}}_{r,i}\|_2^2\right] \\
&= -\mathbb{E}_{\mathcal{Q}(\boldsymbol{Z}_{1:L}, \tilde{\boldsymbol{Y}}_{1:R}|\boldsymbol{x})}\left[\frac{1}{2\sum_{l \in \ell_r} s_l^2}\|\tilde{\boldsymbol{y}}_{r,i} - \boldsymbol{y}_{r,i}\|_2^2\right] + \frac{d_r}{2}.
\end{aligned}
\tag{27}
$$

Here, we used $\tilde{\boldsymbol{Y}}_r = \hat{\boldsymbol{Y}}_r$ instead of sampling it in a practical implementation.

Table 5: Impact of decay factor $r$ in the exponential schedule defined in Appendix E on reconstruction of images in CIFAR10. RMSE ($\times 10^2$) is evaluated using the test set. Too large decay factor deteriorates the reconstruction performance.

| Model | $r = 1 \times 10^{-6}$ | $r = 3 \times 10^{-6}$ | $r = 1 \times 10^{-5}$ | $r = 3 \times 10^{-5}$ | $r = 1 \times 10^{-4}$ | $r = 3 \times 10^{-4}$ |
|---|---|---|---|---|---|---|
| SQ-VAE-2 | $5.223 \pm 0.016$ | $5.219 \pm 0.017$ | $5.264 \pm 0.016$ | $5.356 \pm 0.016$ | $7.234 \pm 0.021$ | $9.083 \pm 0.019$ |
| RSQ-VAE | $6.544 \pm 0.018$ | $6.149 \pm 0.023$ | $5.697 \pm 0.022$ | $5.579 \pm 0.015$ | $5.802 \pm 0.020$ | $6.439 \pm 0.019$ |

## E  Gumbel-softmax approximation

Since HQ-VAE has discrete latent space, our objective functions (6), (14), and (26) involve expectations with respective to the categorical variables $\boldsymbol{Z}_{1:L}$ (in $\mathcal{Q}(\boldsymbol{Z}_{1:L}, \tilde{\boldsymbol{Z}}_{1:L}|\boldsymbol{x})$). As a common method to make the sampling process of the categorical variables reparameterizable, we replace the categorical distribution (2) with Gumbel-softmax alternatives. In a Gumbel-softmax distribution, a temperature parameter $\tau \in (0, \infty)$ is introduced. The sampling process from a Gumbel-softmax distribution is equivalent to that from a categorical distribution as $\tau \to 0$.

There is a trade-off with the temperature $\tau$. Lower temperatures lead to sampling that closely resembles the original categorical distribution but with larger gradient variances. Conversely, higher temperatures result in greater discrepancies between the categorical and Gumbel-softmax distributions but with smaller gradient variances. It has been shown that annealing the temperature from high to low values during model training performs well in practice (Jang et al., 2017; Sønderby et al., 2017; Williams et al., 2020). Specifically, Jang et al. (2017) proposed an exponential scheduling, $\tau = \max\{c_\tau, \exp(-rt)\}$, where $t$ denotes the global training step. Following the approach of Takida et al. (2022b), we adopt the same annealing schedule with $c_\tau = 0$ and $r = 10^{-5}$ for all the experiments (except for Table 5).

Here, we investigate the sensitivity of the choice of the decay factor $r$. We train two-layer SQ-VAE-2 and RSQ-VAE with codebook capacities of $(d_l, K_l) = (4^2, 512), (8^2, 512)$ and $(d_l, K_l) = (8^2, 32), (8^2, 32)$, respectively. We sweep $r$ while keeping the other settings in Table 5. According to the table, a decay factor that is too large, leading to fast convergence to the categorical distribution, deteriorates the reconstruction performance.

## F  Experimental details

This appendix describes the details of the experiments[2] discussed in Section 5. For all the experiments except for the ones of RSQ-VAE and RQ-VAE on FFHQ and UrbanSound8K in Section 5.2, we construct architectures for the *bottom-up* and *top-down* paths as is described in Figures 1 and 7. To build these paths, we use two common blocks, the Resblock and Convblock by following Child (2021) in Figure 7a; these blocks are shown in Figures 7b and 7c. Here, we denote the width and height of $\boldsymbol{H}_\phi^r(\boldsymbol{x})$ as $w_r$ and $h_r$, respectively, i.e., $\boldsymbol{H}_\phi^r(\boldsymbol{x}) \in \mathbb{R}^{d_b \times w_r \times h_r}$. We set $c_{\mathrm{mid}} = 0.5$ in Figure 7. For all the experiments, we use the Adam optimizer with $\beta_1 = 0.9$ and $\beta_2 = 0.9$. Unless otherwise noted, we reduce the learning rate in half if the validation loss does not improve in the last three epochs.

In HQ-VAE, we deal with the decoder variance $\sigma^2$ by using the update scheme with the maximum likelihood estimation (Takida et al., 2022a). We gradually reduce the temperature parameter of the Gumbel-softmax trick with a standard schedule $\tau = \exp(10^{-5} \cdot t)$ (Jang et al., 2017), where $t$ is the iteration step.

We set the hyperparameters of VQ-VAE to the standard values: the balancing parameter $\beta$ in Equations (7) and (15) is set to 0.25, and the weight decay in EMA for the codebook update is set to 0.99.

Below, we summarize the datasets used in the experiments described in Section 5 below.

---

[2]The source code is attached in the supplementary material.

Table 6: Notation for the convolutional layers in Figure 7.

| Notation | Description |
|---|---|
| $\mathrm{Conv}_d^{(1\times1)}$ | 2D Convolutional layer (channel= $n$, kernel= $1 \times 1$, stride= 1, padding= 0) |
| $\mathrm{Conv}_d^{(3\times3)}$ | 2D Convolutional layer (channel= $n$, kernel= $3 \times 3$, stride= 1, padding= 1) |
| $\mathrm{Conv}_d^{(4\times4)}$ | 2D Convolutional layer (channel= $n$, kernel= $4 \times 4$, stride= 2, padding= 1) |
| $\mathrm{ConvT}_d^{(3\times3)}$ | 2D Transpose convolutional layer (channel= $n$, kernel= $3 \times 3$, stride= 1, padding= 1) |
| $\mathrm{ConvT}_d^{(4\times4)}$ | 2D Transpose convolutional layer (channel= $n$, kernel= $4 \times 4$, stride= 2, padding= 1) |

**CIFAR10.** CIFAR10 (Krizhevsky et al., 2009) contains ten classes of $32 \times 32$ color images, which are separated into 50,000 and 10,000 samples for the training and test sets, respectively. We use the default split and further randomly select 10,000 samples from the training set to prepare the validation set.

**CelebA-HQ.** CelebA-HQ (Karras et al., 2018) contains 30,000 high-resolution face images that are selected from the CelebA dataset by following Karras et al. (2018). We use the default training/validation/test split (24,183/2,993/2,824 samples). We preprocess the images by cropping and resizing them to $256 \times 256$.

**FFHQ.** FFHQ (Karras et al., 2019) contains 70,000 high-resolution face images. In Section 5.1, we split the images into three sets: training (60,000 samples), validation (5,000 samples), and test (5,000 samples) sets. We crop and resize them to $1024 \times 1024$. In Section 5.2, we follow the same preprocessing as in Lee et al. (2022a), wherein the images are split training (60,000 samples) and validation (10,000 samples) sets and they are cropped and resized them to $256 \times 256$.

**ImageNet.** ImageNet (Deng et al., 2009) contains 1000 classes of natural images in RGB scales. We use the default training/validation/test split (1,281,167/50,000/100,000 samples). We crop and resize the images to $256 \times 256$.

**UrbanSound8K.** UrbanSound8K (Salamon et al., 2014) contains 8,732 labeled audio clips of urban sound in ten classes, such as dogs barking and drilling sounds. UrbanSound8K is divided into ten folds, and we use the folds 1-8/9/10 as the training/validation/test split. The duration of each audio clip is less than 4 seconds. In our experiments, to align the lengths of input audio, we pad all the audio clips to 4 seconds. We also convert the clips to 16 bit and down-sampled them to 22,050 kHz. The 4-second waveform audio clip is converted to a Mel spectrogram with shape $80 \times 344$. We preprocess each audio clip by using the method described in the paper (Liu et al., 2021):

1. We extract an 80-dimensional Mel spectrogram by using a short-time Fourier transform (STFT) with a frame size of 1024, a hop size of 256, and a Hann window.
2. We apply dynamic range compression to the Mel spectrogram by first clipping it to a minimum value of $1 \times 10^{-5}$ and then applying a logarithmic transformation.

### F.1 SQ-VAE-2 vs VQ-VAE-2

### F.1.1 Comparison on CIFAR10 and CelebA-HQ

We construct the architecture as depicted in Figures 1 and 7. To build the *top-down* paths, we use two *injected top-down* layers (i.e., $R = 2$) with $w_1 = h_1 = 8$ and $w_2 = h_2 = 16$ for CIFAR10, and three layers (i.e., $R = 3$) with $w_1 = h_1 = 8$, $w_2 = h_2 = 16$ and $w_3 = h_3 = 32$ for CelebA-HQ. For the *bottom-up* paths, we repeatedly stack two Resblocks and an average pooling layer once and four times, respectively, for CIFAR10 and CelebA-HQ. We set the learning rate to 0.001 and train all the models for a maximum of 100 epochs with a mini-batch size of 32. The sensitivity of SQ-VAE-2 in terms of batch size is investigated in Figure 8a.

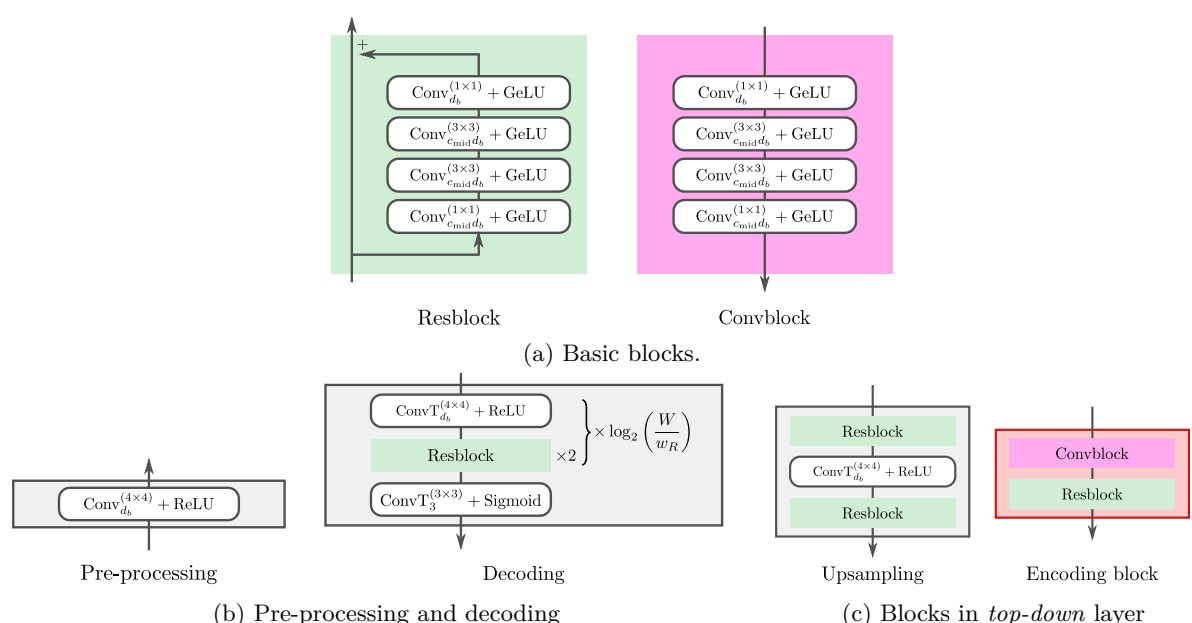

Figure 7: Architecture details in Figure 1. Table 6 summarizes the notation for the convolutional layers, $\mathrm{Conv}_d^{(k \times k)}$ and $\mathrm{ConvT}_d^{(k \times k)}$.

### F.1.2 Comparison on large-scale datasets

We construct the architecture as depicted in Figures 1 and 7. To build the *top-down* paths, we use two *injected top-down* layers (i.e., $R = 2$), with $w_1 = h_1 = 32$ and $w_2 = h_2 = 64$ for ImageNet, and three layers (i.e., $R = 3$) with $w_1 = h_1 = 32$, $w_2 = h_2 = 64$ and $w_3 = h_3 = 128$ for FFHQ, respectively. For the *bottom-up* paths, we repeatedly stack two Resblocks and an average pooling layer three times and five times respectively for ImageNet and FFHQ. We set the learning rate to 0.0005. We train ImageNet and FFHQ for a maximum of 50 and 200 epochs with a mini-batch size of 512 and 128, respectively. Figure 9 and Figure 10 show reconstructed samples of SQ-VAE-2 on ImageNet and FFHQ.

### F.2 RSQ-VAE vs RQ-VAE

### F.2.1 Comparison on CIFAR10 and CelebA-HQ

We construct the architecture as depicted in Figures 1 and 7 without *injected top-down* layers, i.e., $R = 1$. We set the resolution of $\boldsymbol{H}_\phi(\boldsymbol{x})$ to $w = h = 8$. For the *bottom-up* paths, we stack two Resblocks on the average pooling layer once for CIFAR10 and four times for CelebA-HQ. We set the learning rate to 0.001 and train all the models for a maximum of 100 epochs with a mini-batch size of 32. The sensitivity of RSQ-VAE in terms of batch size is investigated in Figure 8b.

### F.2.2 Improvement in perceptual quality

This experiment use the same network architecture as in Lee et al. (2022a). We set the learning rate to 0.001 and train an RSQ-VAE model for a maximum of 300 epochs with a mini-batch size of 128 (4 GPUs, 32 samples for each GPU) on FFHQ. We use our modified LPIPS loss (see Appendix F.5) in the training. For the evaluation, we compute rFID scores with the code provided in their repository[3] on the validation set (10,000 samples). Moreover, we use the pre-trained RQ-VAE model offered in the same repository for evaluating RQ-VAE.

We will show examples of reconstructed images in Appendix F.5 after we explain our modified LPIPS loss.

---

[3]https://github.com/kakaobrain/rq-vae-transformer

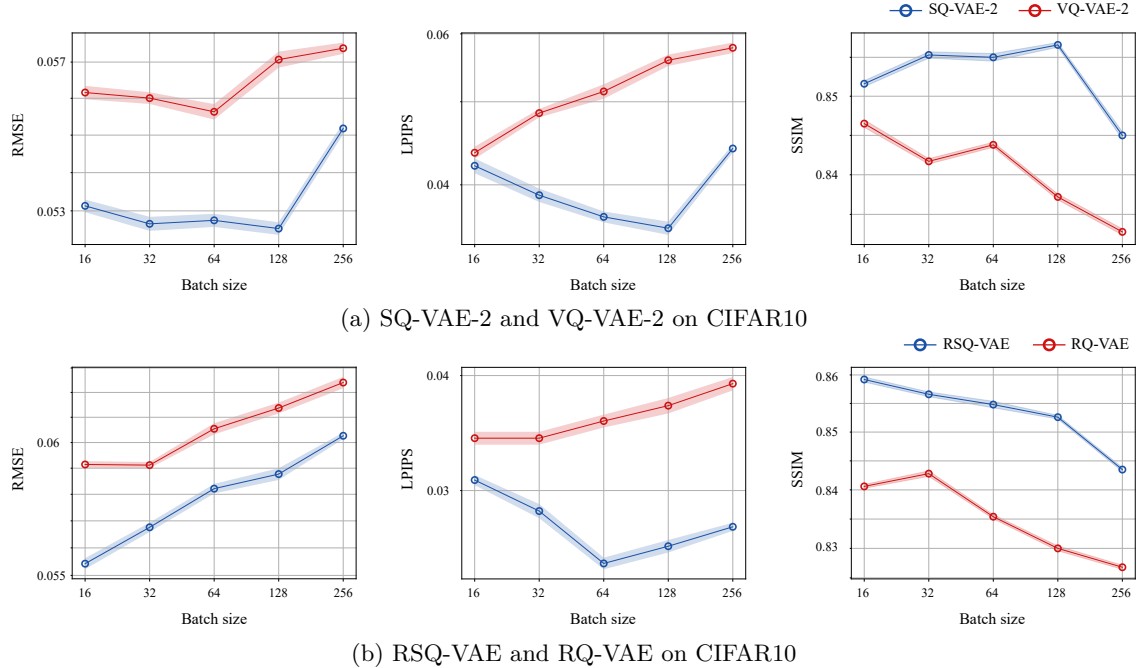

(a) SQ-VAE-2 and VQ-VAE-2 on CIFAR10

(b) RSQ-VAE and RQ-VAE on CIFAR10

Figure 8: Impact of batch size on reconstruction of images in CIFAR10. (a) We set the codebook capacity for the discrete space to $(d_l, K_l) = (4^2, 512), (8^2, 512)$ with $L = 2$. (b) We set the codebook capacity for the discrete space to $(d_l, K_l) = (8^2, 32), (8^2, 32)$ with $L = 2$. For (a) and (b), we use the same architectures as in Sections 5.1 and 5.2, respectively. The plots suggest that batch size should be tuned to achieve better reconstruction performance for all the models.

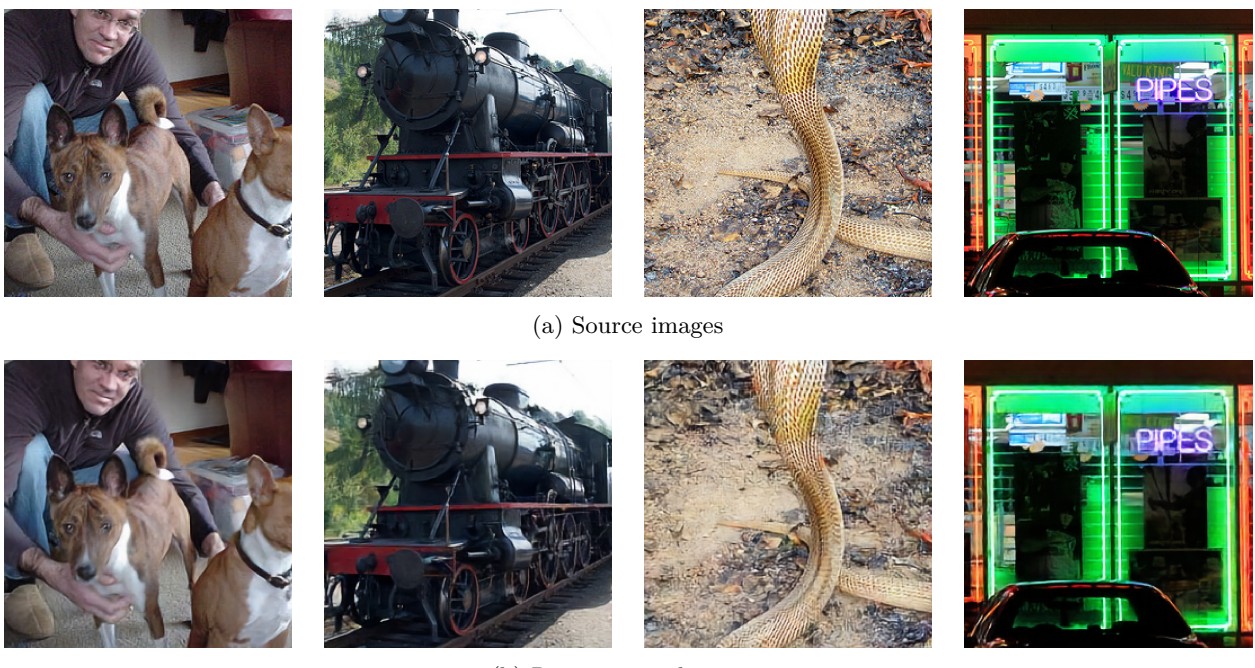

(a) Source images

(b) Reconstructed images

Figure 9: Reconstructed samples of SQ-VAE-2 trained on ImageNet

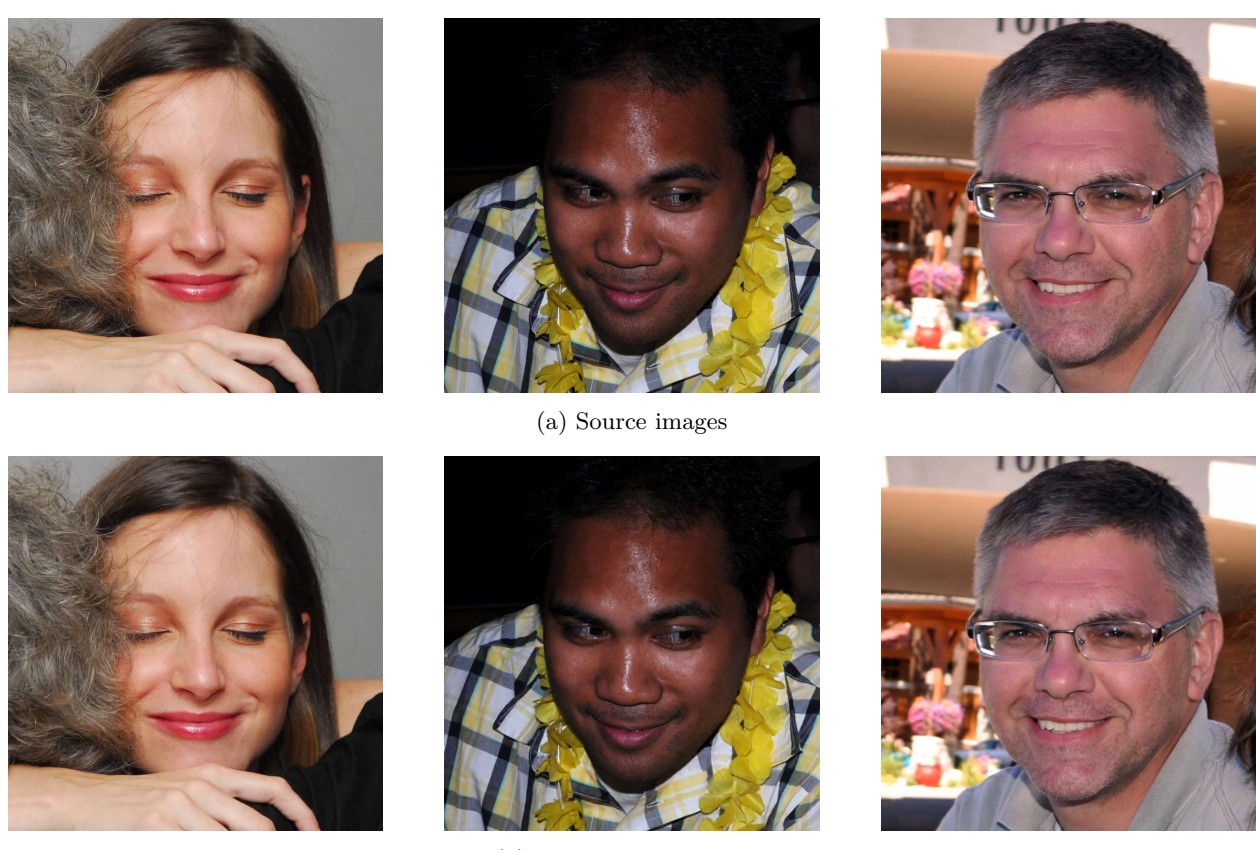

(a) Source images

(b) Reconstructed images

Figure 10: Reconstructed samples of SQ-VAE-2 trained on FFHQ

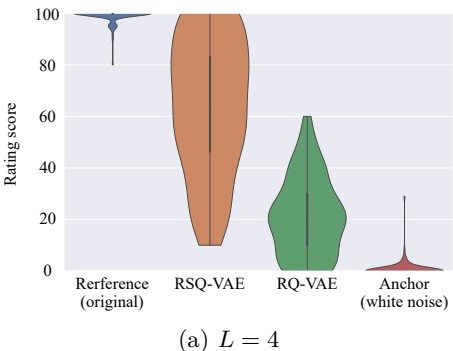
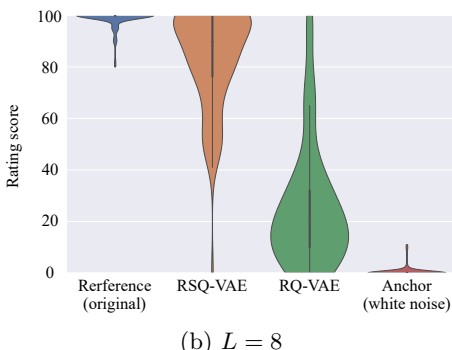

(a) $L = 4$                                           (b) $L = 8$

Figure 11: Violin plots of MUSHRA listening test results on UrbanSound8K test set in cases of (a) four layers and (b) eight layers. The white dots indicate the median scores, and the tops and bottoms of the thick vertical lines indicate the first and third quartiles, respectively.

### F.2.3 Validation on an audio dataset

We construct the architecture in accordance with the previous paper on audio generation (Liu et al., 2021). For the *top-down* paths, the architecture consists of several strided convolutional layers in parallel (Xian et al., 2021). We use four strided convolutional layers consisting of two sub-layers with stride 2, followed by two ResBlocks with ReLU activations. The kernel sizes of these four strided convolutional layers are $2 \times 2$, $4 \times 4$, $6 \times 6$ and $8 \times 8$ respectively. We add the outputs of the four strided convolutional layers together and pass the result to a convolutional layer with a kernel size of $3 \times 3$. Then, we get the resolution of $\boldsymbol{H}_\phi(\boldsymbol{x})$ to $w = 20, h = 86$. For the *bottom-up* paths, we stack a convolutional layer with a kernel size $3 \times 3$, two Resblocks with ReLU activations, and two transposed convolutional layers with stride 2 and kernel size $4 \times 4$. We set the learning rate to 0.001 and train all the models for a maximum of 100 epochs with a mini-batch size of 32.

To evaluate the perceptual quality of our results, we perform a subjective listening test using the multiple stimulus hidden reference anchor (MUSHRA) protocol (Series, 2014) on an audio web evaluation tool (Schoeffler et al., 2018). We randomly select an audio signal from the UrbanSound8K test set for the practicing part. In the test part, we extract ten samples by randomly selecting an audio signal per class from the test set. We prepare four samples for each signal: reconstructed samples from RQ-VAE and RSQ-VAE, a white noise signal as a hidden anchor, and an original sample as a hidden reference. Because RQ-VAE and RSQ-VAE are applied to the normalized log-Mel spectrograms, we use the same HiFi-GAN vocoder (Kong et al., 2020) as used in the audio generation paper (Liu et al., 2021) to convert the reconstructed spectrograms to the waveform samples. The HiFi-GAN vocoder is trained on the training set of UrbanSound8K from scratch (Liu et al., 2021). As the upper bound of quality of the reconstructed waveform is limited to the vocoder result of the original spectrogram, we use the vocoder result for the reference. After listening to the reference, assessors are asked to rate the four different samples according to their similarity to the reference on a scale of 0 to 100. The post-processing of the assessors followed the paper (Series, 2014): assessors were excluded from the aggregated scores if they rated the hidden reference for more than 15% of the test signals with a score lower than 90. After the post-screening of assessors (Series, 2014), a total of ten assessors participated in the test. Figure 11 shows the violin plots of the listening test. RSQ-VAE achieves better median listening scores comapared with RQ-VAE. The scores of RSQ-VAE are better by a large margin especially in the case of eight layers.

As a demonstration, we randomly select audio clips from our test split of UrbanSound8K and show their reconstructed Mel spectrogram samples from RQ-VAE and RSQ-VAE in Figure 13. While the samples from RQ-VAE have difficulty in reconstructing the sources with shared codebooks, the samples from RSQ-VAE reconstruct the detailed features of the sources.

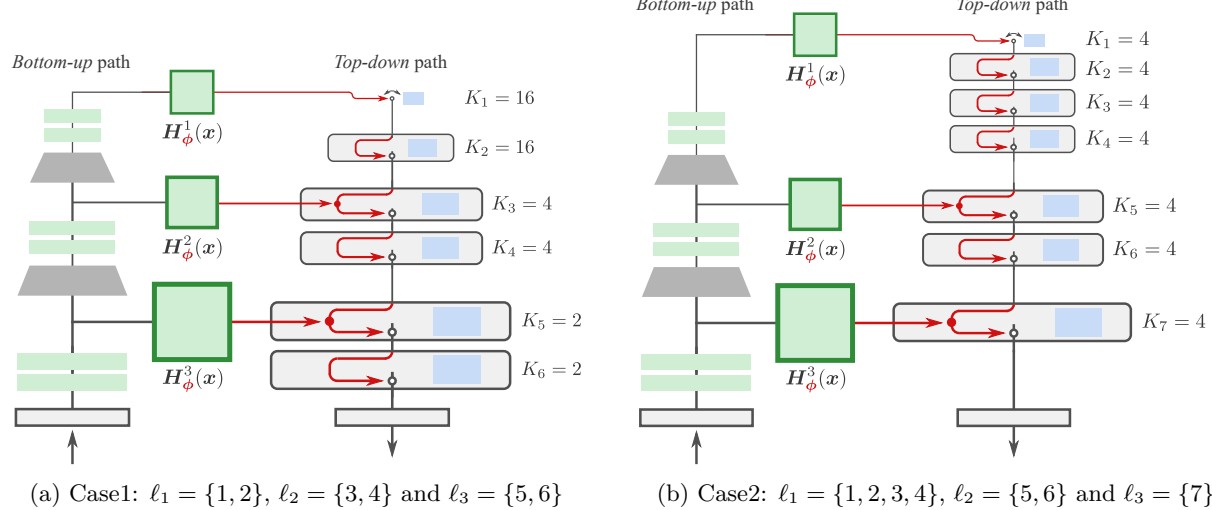

(a) Case1: $\ell_1 = \{1, 2\}$, $\ell_2 = \{3, 4\}$ and $\ell_3 = \{5, 6\}$      (b) Case2: $\ell_1 = \{1, 2, 3, 4\}$, $\ell_2 = \{5, 6\}$ and $\ell_3 = \{7\}$

Figure 12: Architecture of the hybrid model in Appendix F.3.

Table 7: Image generation on FFHQ (a detailed version of Table 3).

| Model | Latent size ($d_l$) | Codebook size ($K_l$) | GAN | rFID↓ | FID↓ |
|---|---|---|---|---|---|
| VQ-GAN (Esser et al., 2021b) | $16^2$ | 1024 | ✓ | - | 11.4 |
| RQ-VAE (Lee et al., 2022a) | $8^2 \times 4$ | $2048 \times 4$ | ✓ | 7.29 | 10.38 |
| RSQ-VAE + RQ-Transformer (ours) | $8^2 \times 4$ | $2048 \times 4$ | | 8.47 | 9.74 |
| RSQ-VAE + Contextual RQ-Transformer (ours) | | | | | 8.46 |

### F.3 Empirical study of top-down layers

As a demonstration, we build two HQ-VAEs by combinatorially using both the *injected top-down* and the *residual top-down* layers with three resolutions, $w_1 = h_1 = 16$, $w_2 = h_2 = 32$ and $w_3 = h_3 = 64$. We construct the architectures as described in Figure 12 and train them on CelebA-HQ. Figure 15 shows the progressively reconstructed images for each case. We can see the same tendencies as in Figures 4a and 4c.

### F.4 Applications of HQ-VAEs to generative tasks

In training RSQ-VAE on FFHQ, we borrow the encoder–decoder architecture from Lee et al. (2022a), which is constructed by adding an encoder and decoder block to VQ-GAN (Esser et al., 2021b) to decrease the resolution of the feature map by half. Table 7 compares our model with other VQ-based generative models, where it can be seem that our model does not rely on the adversarial training framework but is still competitive with RQ-VAE, which is trained with a PatchGAN (Isola et al., 2017). If we evaluate whole pipelines including the VAE and prior model parts as generative models, our models outperform the baseline model. Figure 16 shows the samples generated from RSQ-VAE with the contextual Transformer.

The encoder–decoder architecture for our SQ-VAE-2 on ImageNet is built in the same manner as is shown in Figures 1 and 7. Our architecture structure most resembles that of VQ-VAE-2 because the latent structure is the same. Because ImageNet with a resolution of 256 is a challenging dataset to compress to discrete representations with feasible latent sizes, we use adversarial training for reconstructed images. Subsequently, we train two Muse (Chang et al., 2023) prior models to approximate $Q(\boldsymbol{Z}_1|c)$ and $Q(\boldsymbol{Z}_2|\boldsymbol{Z}_1, c)$, where $c \in [1000]$ indicates the class label. Table 8 compares our model with other VQ-based generative models without the technique of rejection sampling in terms of various metrics. As summarized in the table, our model is competitive with the current state-of-the-art models. Figure 17 shows the generated samples from SQ-VAE-2 with the Muse.

Table 8: Image generation on ImageNet (a detailed version of Table 4).

| Model | Params | Latent size $(d_l)$ | ♯ of codes $(K_l)$ | GAN | rFID↓ | FID↓ | IS↑ |
|---|---|---|---|---|---|---|---|
| VQ-VAE-2 (Razavi et al., 2019) | 13.5B | $32^2, 64^2$ | 512, 512 | | - | $\sim 31$ | $\sim 45$ |
| DALL-E (Ramesh et al., 2021) | 12B | $32^2$ | 8192 | | - | 32.01 | - |
| VQ-GAN (Esser et al., 2021b) | 1.4B | $16^2$ | 16384 | ✓ | 4.98 | 15.78 | 78.3 |
| VQ-Diffusion (Gu et al., 2022) | 370M | $16^2$ | 16384 | ✓ | 4.98 | 11.89 | - |
| MaskGIT (Chang et al., 2022) | 228M | $16^2$ | 1024 | ✓ | 2.28 | 6.18 | 182.1 |
| ViT-VQGAN (Yu et al., 2022) | 1.6B | $32^2$ | 8192 | ✓ | 1.99 | 4.17 | 175.1 |
| RQ-VAE (Lee et al., 2022a) | 1.4B | $8^2 \times 4$ | $16384 \times 4$ | ✓ | 3.20 | 8.71 | 119.0 |
| RQ-VAE (Lee et al., 2022a) | 3.8B | | | | | 7.55 | 134.0 |
| Contextual RQ-Transformer (Lee et al., 2022b) | 1.4B | $8^2 \times 4$ | $16384 \times 4$ | ✓ | 3.20 | 3.41 | 224.6 |
| HQ-TVAE (You et al., 2022) | 1.4B | $8^2, 16^2$ | 8192, 8192 | ✓ | 2.61 | 7.15 | - |
| Efficient-VQGAN (Cao et al., 2023) | - | $16^2$ | 1024 | ✓ | 2.34 | 9.92 | 82.2 |
| Efficient-VQGAN (Cao et al., 2023) | N/A | $32^2$ | 1024 | ✓ | 0.95 | - | - |
| SQ-VAE-2 (ours) | 1.2B+0.56B | $16^2, 32^2$ | 2048, 1024 | ✓ | 1.73 | 4.51 | 276.81 |

### F.5 Perceptual loss for images

We found that the LPIPS loss (Zhang et al., 2018), which is a perceptual loss for images (Johnson et al., 2016), works well with our HQ-VAE. However, we also noticed that simply replacing $\|\boldsymbol{x} - \boldsymbol{f_\theta}(\boldsymbol{Z}_{1:L})\|_2^2$ in the objective function of HQ-VAE (Equations (6) and (14)) with an LPIPS loss $\mathcal{L}_{\text{LPIPS}}(\boldsymbol{x}, \boldsymbol{f_\theta}(\boldsymbol{Z}_{1:L}))$ leads to artifacts appearing in the generated images. We hypothesize that these artifacts are caused by the max-pooling layers in the VGGNet used in LPIPS. Signals from VGGNet might not reach all of the pixels in backpropagation due to the max-pooling layers. To mitigate this issue, we applied a padding-and-trimming operation to both the generated image $\boldsymbol{f_\theta}(\boldsymbol{Z}_{1:L})$ and the corresponding reference image $\boldsymbol{x}$ before the LPIPS loss function. That is $\mathcal{L}_{\text{LPIPS}}(\text{pt}\,[\boldsymbol{x}], \text{pt}\,[\boldsymbol{f_\theta}(\boldsymbol{Z}_{1:L})])$, where pt [ ] denotes our padding-and-trimming operator. The PyTorch implementation of this operation is described below.

```python
import random
import torch
import torch.nn.functional as F

def padding_and_trimming(
    x_rec, # decoder output
    x  # reference image
):
    _, _, H, W = x.size()

    x_rec = F.pad(x_rec, (15, 15, 15, 15), mode='replicate')
    x = F.pad(x, (15, 15, 15, 15), mode='replicate')

    _, _, H_pad, W_pad = x.size()
    top = random.randrange(0, 16)
    bottom = H_pad - random.randrange(0, 16)
    left = random.randrange(0, 16)
    right = W_pad - random.randrange(0, 16)

    x_rec = F.interpolate(x_rec[:, :, top:bottom, left:right],
                    size=(H, W), mode='bicubic', align_corners=False)
    x = F.interpolate(x[:, :, top:bottom, left:right],
                size=(H, W), mode='bicubic', align_corners=False)

    return x_rec, x
```

Note that our padding-and-trimming operation includes downsampling with a random ratio. We assume that this random downsampling provides a generative model with diversified signals in backpropagation across training iterations, which makes the model more generalizable.

Figure 14 shows images reconstructed by an RSQ-VAE model trained with a normal LPIPS loss, $\mathcal{L}_{\text{LPIPS}}(\boldsymbol{x}, \boldsymbol{f_\theta}(\boldsymbol{Z}_{1:L}))$, and ones reconstructed by an RSQ-VAE model trained with our modified LPIPS loss, $\mathcal{L}_{\text{LPIPS}}(\text{pt}[\boldsymbol{x}], \text{pt}[\boldsymbol{f_\theta}(\boldsymbol{Z}_{1:L})])$. As shown, our padding-and-trimming technique alleviates the artifacts issue. For example, vertical line noise can be seen in the hairs in the images generated by the former model, but those lines are removed from or softened in the images generated by the latter model. Indeed, our technique improves rFID from 10.07 to 8.47.

## F.6 Progressive coding

For demonstration purposes in Figure 4a, we incorporate the concept of *progressive coding* (Ho et al., 2020; Shu & Ermon, 2022) into our framework, which helps hierarchical models to be more sophisticated in progressive lossy compression and may lead to high-fidelity samples being generated. Here, one can train SQ-VAE-2 to achieve progressive lossy compression (as in Figure 4a) by introducing additional generative processes $\tilde{\boldsymbol{x}}_l \sim \mathcal{N}(\tilde{\boldsymbol{x}}_l; \boldsymbol{f_\theta}(\boldsymbol{Z}_{1:l}), \sigma_l^2 \boldsymbol{I})$ for $l \in [L]$. We here derive the corresponding ELBO objective with this concept. A more reasonable reconstruction can be obtained from only the higher layers (i.e., using only low-resolution information $\boldsymbol{H}_\phi^r(\boldsymbol{x})$).

We consider corrupted data $\tilde{\boldsymbol{x}}_l$ for $l \in [L]$, which can be obtained by adding noise, for example, $\tilde{\boldsymbol{x}}_l = \boldsymbol{x} + \boldsymbol{\epsilon}_l$. We here adopt the Gaussian distribution $\epsilon_{l,d} \sim \mathcal{N}(0, v_l)$ for the noises. Note that $\{\sigma_l^2\}_{l=1}^L$ is set to be a non-increasing sequence. We model the generative process by using only the top $l$ groups, as $p_{\boldsymbol{\theta}}^l(\tilde{\boldsymbol{x}}_l) = \mathcal{N}(\tilde{\boldsymbol{x}}_l; f_{\boldsymbol{\theta}}(\boldsymbol{Z}_{1:l}), \sigma_l^2 \boldsymbol{I})$. The ELBO is obtained as

$$\mathcal{J}_{\text{SQ-VAE-2}}^{\text{prog}} = \sum_{l=1}^L \frac{D}{2} \log \sigma_l^2 + \mathbb{E}_{\mathcal{Q}(\boldsymbol{Z}_{1:L}, \tilde{\boldsymbol{z}}_{1:L}|\boldsymbol{x})} \left[ \frac{\|\boldsymbol{x} - f_{\boldsymbol{\theta}}(\boldsymbol{Z}_{1:l}) + Dv_l\|_2^2}{2\sigma_l^2} + \frac{\|\tilde{\boldsymbol{Z}}_l - \boldsymbol{Z}_l\|_F^2}{2s_l^2} - H(\hat{P}_{s_l^2}(\boldsymbol{Z}_l|\tilde{\boldsymbol{Z}}_l)) \right]. \tag{28}$$

In Section 5.3, we simply set $v_l = 0$ in the above objective when this technique is activated.

This concept can be also applied to the hybrid model derived in Appendix D by considering additional generative processes $p_{\boldsymbol{\theta}}^r(\tilde{\boldsymbol{x}}_r) = \mathcal{N}(\tilde{\boldsymbol{x}}_r; f_{\boldsymbol{\theta}}(\boldsymbol{Z}_{1:L_r}), \sigma_r^2 \boldsymbol{I})$. The ELBO objective is as follows:

$$\mathcal{J}_{\text{HQ-VAE}}^{\text{prog}} = \sum_{r=1}^R \frac{D}{2} \log \sigma_r^2$$

$$+ \mathbb{E}_{\mathcal{Q}(\boldsymbol{Z}_{1:L}, \tilde{\boldsymbol{Y}}_{1:R}|\boldsymbol{x})} \left[ \sum_{r=1}^R \frac{\|\boldsymbol{x} - \boldsymbol{f_\theta}(\boldsymbol{Z}_{1:L_r}) + Dv_r\|_2^2}{2\sigma_r^2} + \sum_{r=1}^R \frac{\left\| \hat{\boldsymbol{Y}}_r - \sum_{l \in \ell_r} \boldsymbol{Z}_l \right\|_F^2}{2\sum_{l \in \ell_r} s_l^2} - \sum_{l=1}^L H(\hat{P}_{s_l^2}(\boldsymbol{Z}_l|\tilde{\boldsymbol{Z}}_l)) \right]. \tag{29}$$

In Section F.3, we simply set $v_l = 0$ in the above objective when this technique is activated.

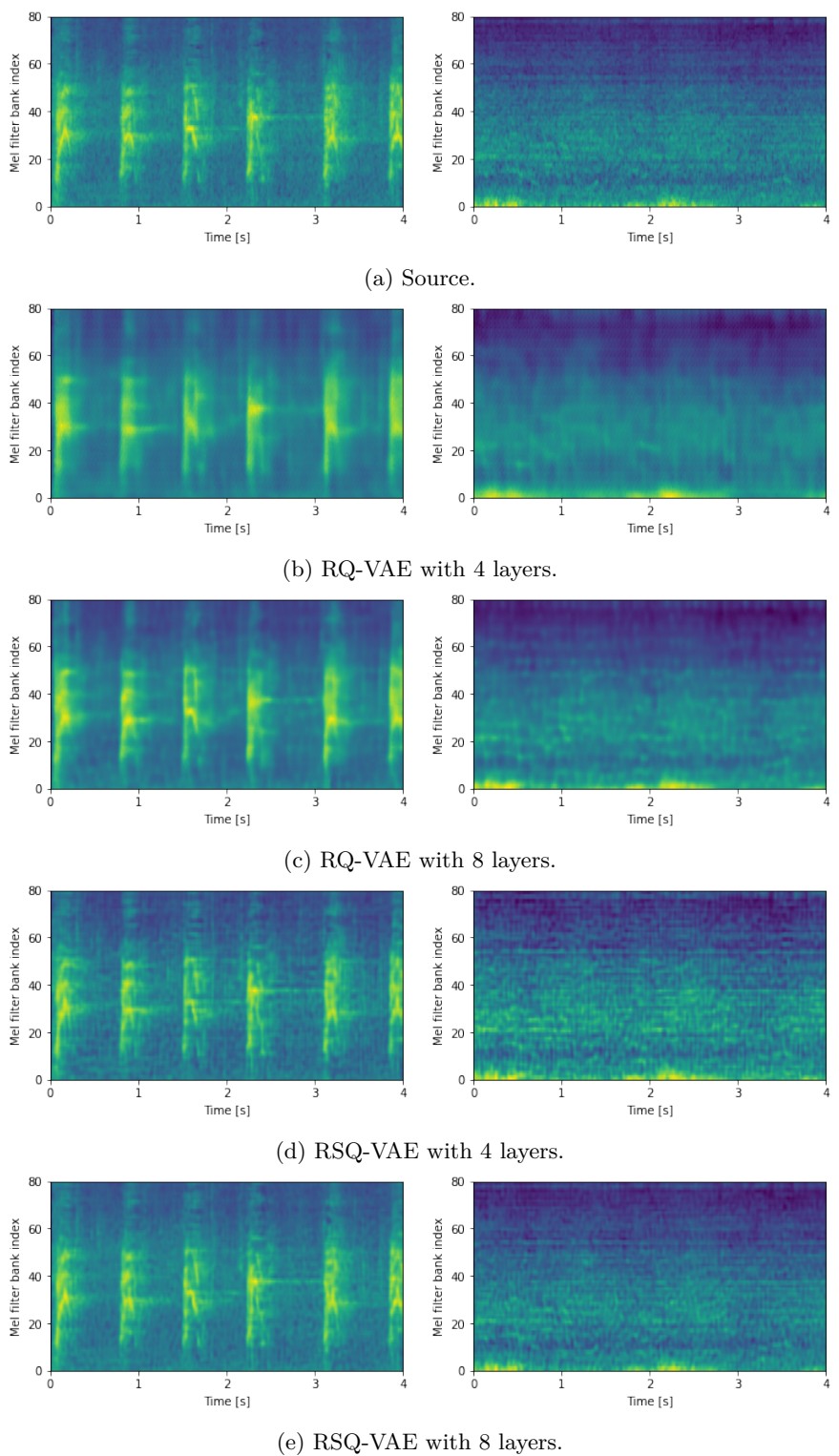

Figure 13: Mel spectrogram of (a) sources and (b)-(e) reconstructed samples of the UrbanSound8K dataset. The left and right panels are audio clips of dog barking and drilling, respectively. We can see that the RQ-VAEs struggle to reconstruct the sources with shared codebooks. In contrast, the reconstruction of RSQ-VAE reflects the details of the source samples.

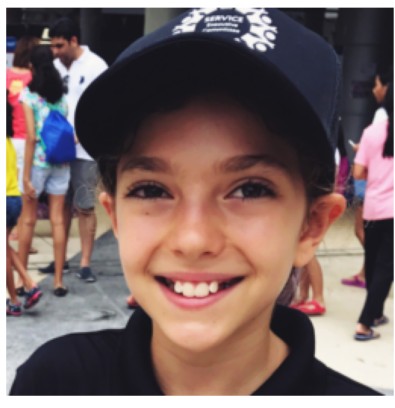 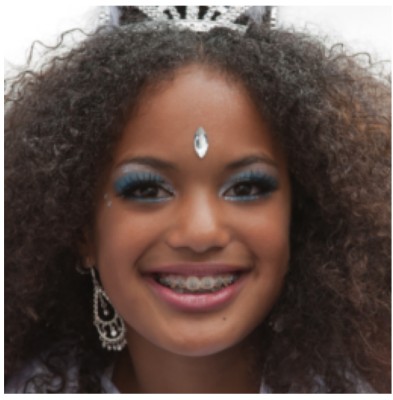 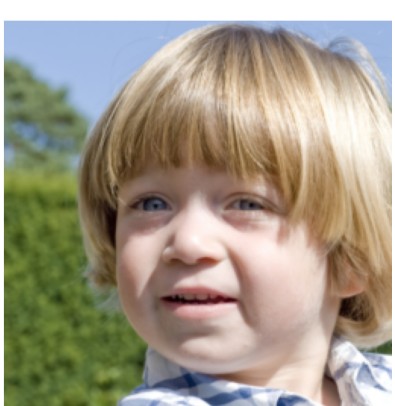

(a) Source

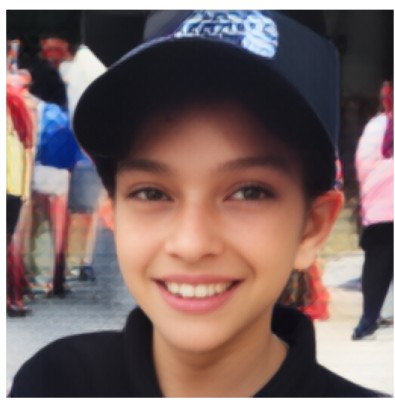 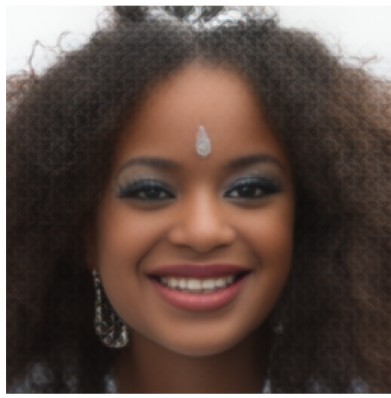 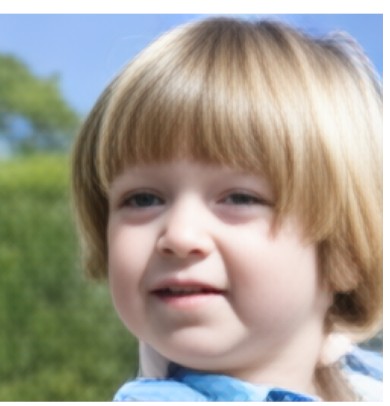

(b) RSQ-VAE trained with a normal LPIPS loss (rFID= 10.07)

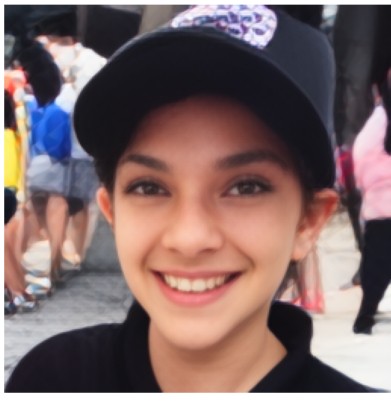 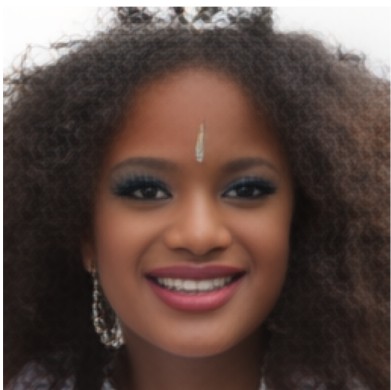 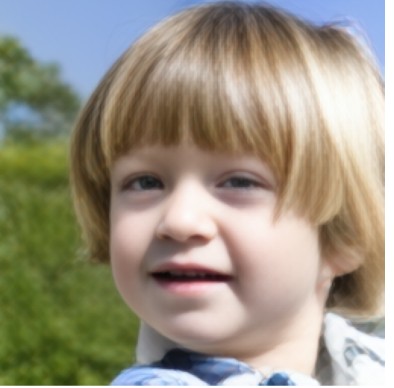

(c) RSQ-VAE trained with our improved LPIPS loss (rFID= 8.47)

Figure 14: Reconstructed samples of FFHQ.

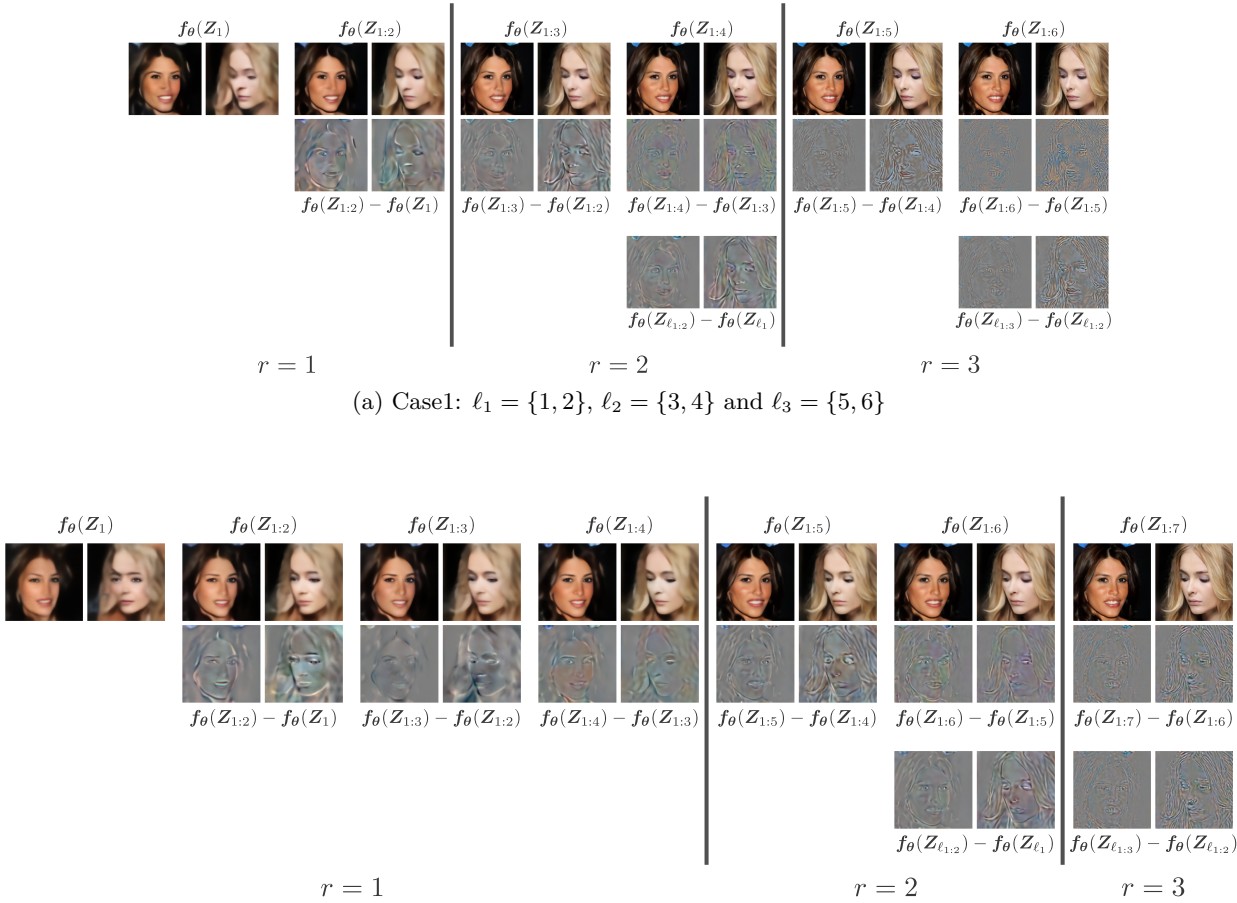

(a) Case1: $\ell_1 = \{1, 2\}$, $\ell_2 = \{3, 4\}$ and $\ell_3 = \{5, 6\}$

(b) Case2: $\ell_1 = \{1, 2, 3, 4\}$, $\ell_2 = \{5, 6\}$ and $\ell_3 = \{7\}$

Figure 15: Reconstructed images and magnified differences of HQ-VAE on CelebA-HQ

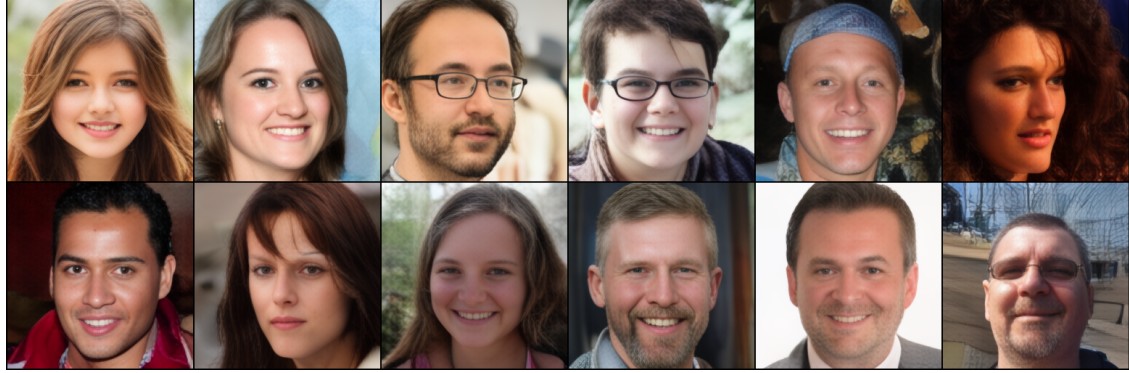

Figure 16: Samples of FFHQ from RSQ-VAE with contextual RQ-Transformer (Lee et al., 2022b).

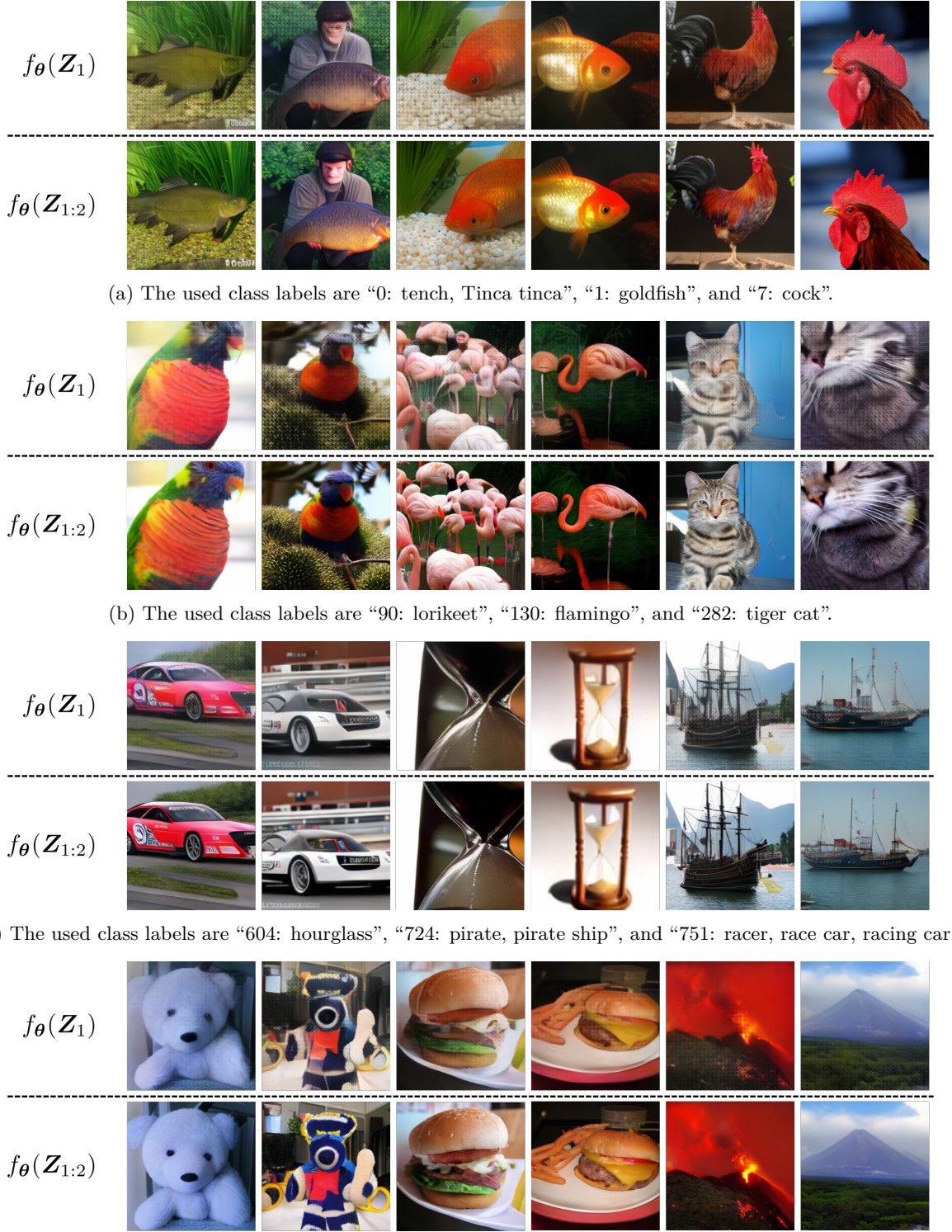

$f_{\boldsymbol{\theta}}(\boldsymbol{Z}_1)$

$f_{\boldsymbol{\theta}}(\boldsymbol{Z}_{1:2})$

(a) The used class labels are "0: tench, Tinca tinca", "1: goldfish", and "7: cock".

$f_{\boldsymbol{\theta}}(\boldsymbol{Z}_1)$

$f_{\boldsymbol{\theta}}(\boldsymbol{Z}_{1:2})$

(b) The used class labels are "90: lorikeet", "130: flamingo", and "282: tiger cat".

$f_{\boldsymbol{\theta}}(\boldsymbol{Z}_1)$

$f_{\boldsymbol{\theta}}(\boldsymbol{Z}_{1:2})$

(c) The used class labels are "604: hourglass", "724: pirate, pirate ship", and "751: racer, race car, racing car".

$f_{\boldsymbol{\theta}}(\boldsymbol{Z}_1)$

$f_{\boldsymbol{\theta}}(\boldsymbol{Z}_{1:2})$

(d) The used class labels are "850: teddy, teddy bear", "933: cheeseburger", and "980: volcano".

Figure 17: Samples of ImageNet from SQ-VAE-2 with Muse (Chang et al., 2023). (Top) Samples generated only with the top layer, $f_{\boldsymbol{\theta}}(\boldsymbol{Z}_1)$. (Bottom) Samples generated with the top and bottom layers, $f_{\boldsymbol{\theta}}(\boldsymbol{Z}_{1:2})$.

