# OpenReview forum: "HQ-VAE: Hierarchical Discrete Representation Learning with Variational Bayes"
_TMLR — Accepted by TMLR_

### Review · Reviewer_dLpy · 2024-01-19

**Summary Of Contributions:**

This paper proposes an extension of the recently proposed SQ-VAE (Takida et al., 2022), a VAE with discrete latents learned through probablistic quantization, into a hierarchical architecture. Overall, the hierarchical VAE architecture is similar to that of ladder VAE with bidrectional inference, but uses the technique of SQ-VAE to learn categorically distributed latents with learned codebooks; two main variants are considered, based on VQ-VAE-2 and residual-quantized VAE. The experiments demonstrate enhanced codebook utilization and higher reconstruction quality on image and audio, and additionally better FID on image generation when pairing the learned latent tokens with a transformer prior.

**Audience:**

Yes

**Claims And Evidence:**

No

**Requested Changes:**

A clearer explanation of how the stochastic quantization in SQ-VAE differs from earlier methods like that of Sønderby et al. (2017) and dVAE, and why this makes a difference, ideally through a small numerical example rather than hand-wavy arguments.

**Strengths And Weaknesses:**

Strengths:

1. This work showed how to extend SQ-VAE into a hierarchical architecture and detailed the design choices that made it work well. This is likely useful for practitioners who work with VQ-VAE type of models and seek an easier-to-use alternative.

2. The paper did a pretty comprehensive evaluation comparing the effect of the SQ-VAE-style categorical posterior & training scheme v.s. the usual VQ-VAE learned quantization scheme, in several popular architecture variants. Although perhaps not surprising, the findings in the SQ-VAE paper (less codebook collapse, better code utilization, etc.) are found to continue to hold in the hierarchical architectures.

Weakness:
It's unclear to me how the stochastic quantization in SQ-VAE differs from earlier methods like that of Sønderby et al. (2017) and dVAE, and what's the advantage of the proposed stochastic quantization if there is indeed a difference. In referring to the earlier methods of Sønderby et al. (2017) and dVAE in Sec 2.2, the paper claimed that "such an index-domain modeling cannot incorporate the codebook geometry explicitly into the posterior modeling. In contrast, VQ/SQ-based methods allow us to model the posterior distribution with vector operations". I'm not sure what it means to "incorporate the codebook geometry explicitly" and don't think the assessment is quite correct, because these earlier methods used something very similar to the stochastic quantization of SQ-VAE; specifically, "For the probabilistic models we let $q(z|x) = Cat(f(x, M))$, where $f(x, M)$ calculates the logits for the categorical distribution using the distance between $v$ and the rows of $M$" (Sønderby et al. (2017)); here $v = f_{enc}(x)$ is some vector representation of the input, and the distances between $v$ and codebook vectors go through a softmax to form the categorical posterior, just like in SQ-VAE.

------- Sønderby, Casper Kaae, Ben Poole, and Andriy Mnih. "Continuous relaxation training of discrete latent variable image models." Beysian DeepLearning workshop, NIPS. Vol. 201. 2017.

Lastly, I know the focus of this work is on VQ, but I'm curious if the authors have any comment on the potential of the scalar quantization approach, especially in light of this paper: [Finite Scalar Quantization: VQ-VAE Made Simple](https://arxiv.org/pdf/2309.15505.pdf); this is a fairly recent paper, so I don't expect any empirical comparisons, and the authors' comment on it won't affect my evaluation of the current submission.

---

> ### Author Response · Authors · 2024-02-16
> **Author reply (1/2)**
>
> We greatly appreciate the valuable feedback you have provided, and have addressed each of your comments. Please kindly find our response below.
>
> **Weakness**
> ---
> > 1. It's unclear to me how the stochastic quantization in SQ-VAE differs from earlier methods like that of Sønderby et al. (2017) and dVAE, and what's the advantage of the proposed stochastic quantization if there is indeed a difference.
>
> We appreciate your comments. We agree that elaborating on the differences helps readers grasp the literature. We would like to explain the differences in the approaches to posterior modeling in SQ-VAE from those in Sønderby et al. (2017), and in dVAE below.
>
> **GS-Soft** proposed in **Sønderby et al. (2017)** is comparable to SQ-VAE. Although the details about the methodology are not available in the original paper, we assume that they modeled the posterior distribution as $Q(\mathbf{z}\_i=\mathbf{b}\_k|\mathbf{x})\propto-||\tilde{\mathbf{z}}\_i-\mathbf{b}\_k||_2^2$ based on several related papers and Github repositories [1,2]. We believe GS-Soft and SQ-VAE have a major difference as follows. In SQ-VAE, they introduced a trainable parameter, denoted as $\mathbf{\Sigma}\_{\varphi}$, by considering a dequantization process as the inverse operation of quantization. This leads to  $$Q(\mathbf{z}\_i=\mathbf{b}\_k|\mathbf{x})\propto-\frac{1}{2}(\tilde{\mathbf{z}}\_i-\mathbf{b}\_k)^\top\mathbf{\Sigma}\_{\varphi}^{-1}(\tilde{\mathbf{z}}\_i-\mathbf{b}\_k).$$
> The parameter brings us several benefits listed below.
>
> (1-1) The trainable parameter induces a self-annealing effect during model training, which is crucial for preventing the codebook collapse issue. In Section 3.3 of the SQ-VAE paper, they provided a theoretical justification for this effect and empirically compared SQ-VAE and GS-Soft to verify it. Since we adopt the same quantization approach, HQ-VAE inherits this property, as observed in SQ-VAE-2 and RSQ-VAE (please refer to Figures 4(a) and 4(c) of our manuscript).
>
> (1-2) The dequantization-based formulation extends the Euclidean distance for posterior modeling to more general distances with parameterization of $\mathbf{\Sigma}\_{\varphi}$ (please refer to Table 1 and Section 3.4 in their paper). It is verified that such a dequantization-based formulation leads to better reconstruction performance in the original SQ-VAE paper. Please note that, for simplicity, we use the most simplified parameterization, i.e., $\sigma^2\mathbf{I}$ for $\mathbf{\Sigma}\_{\varphi}$ (corresponding type-(I) in their paper), in our work.
>
> [1] Williams et al., “Hierarchical Quantized Autoencoders,” NeurIPS, 2020.
>
> [2] https://github.com/bshall/VectorQuantizedVAE
>
> ---
>
> The logits modeling (**LM**) in **dVAE** is another approach to model the discrete posterior distribution ($Q(\mathbf{Z}|\mathbf{x})$) instead of vector quantization. LM assumes an encoder that directly maps from $\mathbf{x}$ to the logits of $Q(\mathbf{z}\_i|\mathbf{x})$. Therefore, LM-based models do not have raw explicit features $\tilde{\mathbf{Z}}$. On the other hand, in SQ-VAE, the encoded features $\tilde{\mathbf{Z}}$ are directly quantized. When the learnable variance parameter $s^2$ is set (fixed) close to $0$, conventional VQ is obtained. This means that SQ-VAE can leverage the strengths of VQ-VAE that LM does not possess.
>
> The benefits of SQ over LM can be listed as follows.
>
> (2-1) SQ can serve as a drop-in replacement of VQ since the required modification only involves the introduction of the learnable parameter $s^2$. This is beneficial because most of the current discrete representation methods published as open-source are based on VQ.
>
> (2-2) With SQ, we can incorporate vector operations into the posterior modeling similarly to VQ. As a representative example, while our paper proposes extending SQ to RQ, LM cannot incorporate such an inductive bias characterized by vector quantization into posterior modeling.
>
> (2-3) With SQ, we can incorporate codebook initialization techniques that are often adopted for improvement in VQ-VAE [3].
>
> In contrast, LM has the following benefit over SQ.
>
> (3-1) LM leads to faster computation for autoencoding since it does not involve the calculation of distances between encoded vectors and all the codebook vectors, the computational complexity of which is $O(K\_l d\_b d\_l)$ for the $l$th layer.
>
> [3] Zheng et al., “Online Clustered Codebook,” ICCV, 2023.

---

> > ### Author Response · Authors · 2024-02-16
> > **Author reply (2/2)**
> >
> > > 2. Lastly, I know the focus of this work is on VQ, but I'm curious if the authors have any comment on the potential of the scalar quantization approach, especially in light of this paper: Finite Scalar Quantization: VQ-VAE Made Simple; this is a fairly recent paper, so I don't expect any empirical comparisons, and the authors' comment on it won't affect my evaluation of the current submission.
> >
> > Thank you for bringing up the interesting question. We believe that finite scalar quantization (FSQ), which does not require codebook training, is another intriguing approach to discrete representation learning. Furthermore, since  the SQ framework is orthogonal to the methodological contributions in FSQ and is applicable to scalars, incorporating them into discrete posterior modeling (for both single-layer and hierarchical representations) should be a future interesting direction. Such a new variational Bayes framework for FSQ may lead to more efficient utilization of scalars and consequently improve reconstruction performance.
> >
> > ---
> >
> > **Requested Changes**
> > ---
> > > A clearer explanation of how the stochastic quantization in SQ-VAE differs from earlier methods like that of Sønderby et al. (2017) and dVAE, and why this makes a difference, ideally through a small numerical example rather than hand-wavy arguments?
> >
> > Thank you for the constructive suggestion. We will modify the original Section 2.2 that explained dVAE to reflect your suggestion as well as Reviewer MV41’s comment as follows. We will review VQ/SQ-based models in the revised Section 2.2 to concentrate on the vector quantization, which is our main focus. Additionally, we will include a new section in the appendix to explain how SQ-VAE differs from the earlier works (please also see our response in the weakness section above ). We believe that adding this section to the appendix will help readers grasp the current literature without distracting them from the main focus.

---

> > > ### Comment · Reviewer_dLpy · 2024-03-10
> > > **Thank you for the response**
> > >
> > > I've read the responses and updated manuscript, which largely addressed my concerns. I believe the improved discussion on how the proposed method relates to existing work is an important contribution, given that the prior SQ-VAE work (Takida et al. 2022) didn't sufficiently address this, so I would recommend placing it the main text (perhaps in the background section, or in a related works section) rather than the appendix.
> > >
> > > ----
> > > Takida et al., 2022. SQ-VAE: Variational Bayes on Discrete Representation with Self-annealed Stochastic Quantization

---

> > > > ### Author Response · Authors · 2024-03-11
> > > > **Author reply**
> > > >
> > > > We greatly appreciate your efforts in reassessing our paper. We are pleased to hear that our responses and revisions effectively addressed your concerns. We will certainly incorporate your additional comments into our revised manuscript.

---

### Review · Reviewer_MV41 · 2024-01-25

**Summary Of Contributions:**

This study presents a new vector quantization method based on a variational autoencoder (VAE). The authors use a variational Bayes framework to develop the hierarchically quantized variational autoencoder (HQ-VAE). HQ-VAE can learn a hierarchical discrete representation is good for efficiently representing data. The authors demonstrate the merits of their approach using image and audio data.

**Audience:**

Yes

**Broader Impact Concerns:**

No concerns

**Claims And Evidence:**

Yes

**Requested Changes:**

Phrasing: P1 “In the original VQ-VAE…” could be rephrased to

Based on the construction of \cite{}, inputs are first encoded and…

P2 “local and global information” these terms are not really defined at this point, so it’s hard to understand what they refer to

The organization of Section 2 could be improved if the subsections were fused and by adding sentences that describe the relation between the methods. Why is the title of dVAE phrased as SQ-VAE vs. dVAE?

P3 “divide up” not sure this is proper phrasing

Please expand the explanation of the method in the caption of figure 1.

P4 “we give” -> we introduce

In eq 9 you use l’ but the summation is over l

Some commas are missing after equations, for example after Eq. 14.

P7 Bad phrasing: “the regularization is too strong to regularize”
Also, the intuition here isn’t clear neither is the solution to this issue.

P8 In the remark you mention that the method does not require any hyperparameters besides the gumbel softmax and point out eq 6 and 14. However, these hyperparameters do not appear there.
Also, isn’t \beta a hyperparameter? How about the learning rate and batch size?

The approximation using the Gumbel softmax is not explained.

**Strengths And Weaknesses:**

Strengths


The authors have tackled an important and challenging problem. This problem has various applications in computer vision and audio processing and is of interest to the research community. The proposed method is novel overall and leads to several improvements compared to existing schemes. The top-down and bottom-up paths seem to help the model hierarchically learn good codebooks. The method is evaluated on several datasets and compared to leading baselines.


Weaknesses

Some aspects of the paper's presentation need improvement, and many parts are hard to follow. Specifically, the English should be improved, and some sections should be reorganized to enhance the paper's readability. I have provided examples for such required changes in the changes bullet.

Some experimental details are missing; for example, is the RMSE evaluated on a left-out test set? If so, what size?
An ablation study is missing, as well as an evaluation of how sensitive the approach is with respect to different changes in parameters.

---

> ### Author Response · Authors · 2024-02-16
> **Author reply (1/2)**
>
> Thank you for your constructive feedback and recognition of our work. We would like to answer your questions and address your concerns below.
>
>
> **Weakness**
> ---
> > 1. Some aspects of the paper's presentation need improvement, and many parts are hard to follow. Specifically, the English should be improved, and some sections should be reorganized to enhance the paper's readability. I have provided examples for such required changes in the changes bullet.
>
> We deeply appreciate your suggestions, which include concrete examples. We will carefully incorporate them into the revised manuscript. Please refer to our reply to each comment in the “Requested Changes” section.
>
> ---
> > 2-1. Some experimental details are missing; for example, is the RMSE evaluated on a left-out test set? If so, what size?
>
> Thank you for your comment. In our experiments, we divided all the datasets into training, validation, and test sets by following common practices, as explained in Appendix D. Throughout the experiments, all the evaluation metrics (e.g., RMSE, LPIPS, SSIM, FID, perplexity) included in the figures and tables were calculated on the test sets, while validation sets were used for scheduling the learning rate. We will explain that all the evaluation metrics are calculated on the test sets in Section 5 and include the sizes of the sets in Appendix D.
>
> ---
> > 2-2. An ablation study is missing, as well as an evaluation of how sensitive the approach is with respect to different changes in parameters.
>
> Thank you for your comment. In response to this feedback, we are now conducting additional experiments to assess the sensitivity of the models in terms of batch size. We would like to note that the parameters regarding the codebooks were already investigated in the original manuscript (please refer to Figures 2 and 3). We would appreciate it if you could let us know of any parameters of interest, in addition to batch size.

---

> > ### Author Response · Authors · 2024-02-16
> > **Author reply (2/2)**
> >
> > **Requested Changes**
> > ---
> > > 1. Phrasing: P1 “In the original VQ-VAE…” could be rephrased to … Based on the construction of \cite{}, inputs are first encoded and…
> >
> > We will modify the sentence according to the suggestion.
> >
> > ---
> > > 2. P2 “local and global information” these terms are not really defined at this point, so it’s hard to understand what they refer to
> >
> > By following [1,2], we will include examples of “local” and “global” information on page 1.
> >
> > [1] Razavi et al., “Generating Diverse High-Fidelity Images with VQ-VAE-2,” NeurIPS, 2019.
> >
> > [2] Dhariwal e al., “Jukebox: A Generative Model for Music,” arXiv, 2020.
> >
> > ---
> > > 3. The organization of Section 2 could be improved if the subsections were fused and by adding sentences that describe the relation between the methods. Why is the title of dVAE phrased as SQ-VAE vs. dVAE?
> >
> > We appreciate your suggestion. We will merge the subsections into one section, reviewing VQ-VAE, VQ-VAE-2, RQ-VAE, SQ-VAE. Additionally, we will add sentences describing the relationship between the models. Please note that, in response to Reviewer dLpy’s comment, we will move explanations about dVAE to the appendix in order to provide a more detailed comparison of SQ-VAE and dVAE.
> >
> > ---
> > > 4. P3 “divide up” not sure this is proper phrasing
> >
> > We will revise the phrase in the manuscript.
> >
> > ---
> > > 5. Please expand the explanation of the method in the caption of figure 1.
> >
> > We will provide further explanation in the caption to help readers better understand our proposal.
> >
> > ---
> > > 6. P4 “we give” -> we introduce
> >
> > We will modify it according to your suggestion.
> >
> > ---
> > > 7. In eq 9 you use l’ but the summation is over l
> >
> > Thanks for the keen observation. The summation should be over $l^\prime$ instead of $l$. We will modify the equation accordingly.
> >
> > ---
> > > 8. Some commas are missing after equations, for example after Eq. 14.
> >
> > In response to your comment, we have checked all the equations throughout the manuscript and found that Equation (28) lacks a period. We will modify Equations (14) and (28) accordingly.
> >
> > ---
> > > 9. P7 Bad phrasing: “the regularization is too strong to regularize” Also, the intuition here isn’t clear neither is the solution to this issue.
> >
> > Thank you for the suggestion. We will modify the sentence to clarify our intention and avoid the awkward phrasing.
> >
> > ---
> > > 10. P8 In the remark you mention that the method does not require any hyperparameters besides the gumbel softmax and point out eq 6 and 14. However, these hyperparameters do not appear there. Also, isn’t \beta a hyperparameter? How about the learning rate and batch size?
> >
> > We appreciate your question. We believe that providing further elaboration on the hyperparameters will highlight the benefits of HQ-VAE over existing VQ-based models. Here, we would like to clarify our intention.
> >
> >
> > VQ-VAE and its variants have three types of hyperparameters: (1) deep learning optimization parameters (e.g., architecture size, batch size, learning rate),  (2) codebook capacity parameters (e.g., $L$, $K/_l$, and $d_l$), and (3) objective function parameters (e.g., $\beta$ and a weight for EMA in codebook learning) in their objective functions.
> >
> > Hyperparameter tuning for (1) is inevitable even in HQ-VAEs as long as common frameworks for training deep learning are used.
> >
> > Regarding (2), the codebook capacity should be set in advance in VQ/SQ-based models. The RD curves obtained from HQ-VAEs (Figures 2 and 3) demonstrate that the reconstruction performance improves with an increase in codebook capacity. The desirable trade-off in HQ-VAEs eliminates the need for additional capacity tuning when a maximum capacity is specified.
> >
> > As for (3), in VQ-VAE models, one has to tune several hyperparameters such as the balancing coefficients (e.g., $\beta$) in their objective functions, as well as those related to some heuristics (e.g., the EMA update only for the codebook and the codebook reset). In contrast, our objective functions (Equations 6 and 14) do not require such hyperparameters.
> >
> > We will elaborate further for this remark.
> >
> > ---
> > > 11. The approximation using the Gumbel softmax is not explained.
> >
> > Thank you for the suggestion. In order to enhance the self-contained nature of our manuscript, we will include a section explaining the Gumbel-softmax approximation in the appendix.

---

> > > ### Comment · Reviewer_MV41 · 2024-03-03
> > > **Response to authors**
> > >
> > > I appreciate the revisions made by the authors as they have done a good job in addressing my concerns. It would be valuable to add an ablation to the study, specifically by reducing the variance of the gumbel noise (making it approximately deterministic) and checking the effect of different elements in Eq. 14. Additionally, as mentioned by the authors, stability for different batch sizes should be assessed.

---

> > > > ### Author Response · Authors · 2024-03-04
> > > > **Author reply**
> > > >
> > > > Thank you very much for taking the time to reply to us and for considering our responses. We sincerely appreciate your efforts in reviewing our revisions. We would be happy to incorporate your additional comment as follows.
> > > >
> > > > > It would be valuable to add an ablation to the study, specifically by reducing the variance of the gumbel noise (making it approximately deterministic) and checking the effect of different elements in Eq. 14.
> > > >
> > > > We appreciate your additional comment. In response to this, we are currently conducting an additional study concerning the Gumbel-softmax approximation, specifically by varying the parameter $r$ as defined in Appendix E. Here, we would like to clarify the effect of the temperature parameter in the Gumbel-softmax approximation.
> > > >
> > > > Our quantization formulation (the same as that in SQ-VAE) involves two types of scalar parameters: (1) the Gumbel temperature, $\tau$ (please also see Appendix E), which **is scheduled** and (2) the dequantization variances, $(s\_{l}^2)\_{l=1}^L$, which **are optimized**. These parameters have different effects on the model training as follows.
> > > >
> > > > The temperature, $\tau$, serves a distinct purpose from controlling quantization stochasticity. We replaced the categorical distributions to be sampled from with Gumbel-softmax alternatives to utilize the reparameterization trick for approximating sampling from the categorical distributions. There is a trade-off with the temperature, which can take on any positive and finite value. Lower temperatures lead to sampling that closely resembles the original categorical distribution but with larger gradient variances. Conversely, higher temperatures result in greater discrepancies between the categorical and Gumbel-softmax distributions but with smaller gradient variances. Thus, a low temperature parameter yields a Gumbel-softmax distribution nearly identical to the original distribution, although this does not necessarily imply approximately deterministic quantization. Annealing the temperature from high to low values during model training has been shown to perform well in practice [1,2,3]. For the temperature annealing, we adopted the exponential scheduling proposed in [1].
> > > >
> > > > To model the stochasticity of the quantization process, we introduce the dequantization variances ($(s\_{l}^2)\_{l=1}^L$) in the latent space. The variances are expected to gradually decrease toward zero along with the data variance ($\sigma^2$), during the training. This means that **the quantization automatically transitions from a nearly random to an almost deterministic process as training progresses**. This phenomenon is referred to as the self-annealing effect. The ELBO-based objective functions naturally induce this as follows: the first and second terms in Equations (6) and (14) induce data reconstruction through the overall autoencoding path, reducing both $\sigma^2$ and $(s\_{l}^2)\_{l=1}^L$; the third and fourth terms regularize the latent space and have the effect of increasing the variances $(s\_{l}^2)\_{l=1}^L$, preventing them from decreasing rapidly. This **gradual** transition from a random to a deterministic quantization process is crucial for mitigating the collapse issue. We observed this phenomenon in HQ-VAE, as depicted in Figures 4 (b,d).
> > > >
> > > > [1] Jang et al., "Categorical reparameterization with gumbel-softmax," ICLR, 2017
> > > >
> > > > [2] Sønderby et al., "Continuous relaxation training of discrete latent variable image models," Beysian DeepLearning workshop, NIPS, 2017
> > > >
> > > > [3] Williams et al., “Hierarchical Quantized Autoencoders,” NeurIPS, 2020.
> > > >
> > > > ---
> > > > > Additionally, as mentioned by the authors, stability for different batch sizes should be assessed.
> > > >
> > > > We included the empirical study on batch size in Figure 8 of the revised manuscript. According to the result, there is no significant difference in the sensitivity of HQ-VAE and VQ-VAE variants with respect to the batch size.

---

> > > > > ### Author Response · Authors · 2024-03-06
> > > > > **Follow-up reply**
> > > > >
> > > > > We appreciate your suggestion regarding the experiments on the Gumbel temperature once again. We have added experimental results demonstrating the sensitivity of the models in terms of the decay factor $r$ used in the exponential schedule for the temperature, as shown in Table 5. Additionally, we have included a paragraph in Appendix E to explain the trade-off with $\tau$.

---

### Review · Reviewer_bvQe · 2024-02-06

**Summary Of Contributions:**

This paper proposes Hierarchically Quantized VAE (HQ-VAE) for learning hierarchical latent representations. The main idea is to use the corresponding bottom-up representation and the top-down representations up to the layer $l-1$ to compute the top-down representation at the layer $l$. Conceptually, this paper is a combination of VQ-VAE-2 for learning hierarchical representations and SQ-VAE for a stochastic matching between the continuous latent code $\tilde{Z}_i$ and discrete codeword vectors.

**Audience:**

Yes

**Broader Impact Concerns:**

There are no broader impact concerns.

**Claims And Evidence:**

Yes

**Requested Changes:**

I have some questions for the authors:

1) What are differences between your approach to learn latent representations and VQ-VAE-2?

2) Section 4.3 needs to be improved for more clarification. What is $H_\phi(x)$? How can you do $\sum_{i=1}^l Z_i$ because they have different dimensions?

3) How to compute  the bottom-up path $H^{r(l)}_{\phi}(x))$? Is it the same as Stage 1: Learning Hierarchical Latent Codes in VQ-VAE?

4) The authors should show the generated images for Table 3.

**Strengths And Weaknesses:**

Strengths
- The paper is generally well-written

- The experiments are sufficiently rich to demonstrate the efficiency of the proposed approach.

Weaknesses
- The novelty is somehow limited. This paper seems to be a combination of VQ-VAE-2 for learning hierarchical representations and SQ-VAE for a stochastic matching between the continuous latent code $\tilde{Z}_i$ and discrete codeword vectors.

---

> ### Author Response · Authors · 2024-02-16
> **Author reply (1/2)**
>
> Thank you for reviewing our paper carefully. Your comments and questions are valuable for improving our work. We have thoroughly considered each of your points and have provided our response below.
>
>
> **Weakness**
> ---
> > The novelty is somehow limited. This paper seems to be a combination of VQ-VAE-2 for learning hierarchical representations and SQ-VAE for a stochastic matching between the continuous latent code $\tilde{Z}\_i$ and discrete codeword vectors.
>
> We would like to clarify the novelty of our work. Our goal is to establish a variational Bayes framework for learning discrete representations using a **generic hierarchical structure**. Specifically, our framework comprises the following contributions in terms of methodology.
>
> (1) We formulated HQ-VAE, a VAE equipped with general hierarchical discrete space, employing a tractable autoencoding architecture that consists of bottom-up and top-down paths. This design facilitates the flexible and straightforward design of the hierarchical structure. The formulation follows variational Bayes, distinguishing it from VQ-VAE and its variants (e.g., VQ-VAE-2 and RQ-VAE).
>
> (2) We provided two instances of HQ-VAE, SQ-VAE-2 and RSQ-VAE, which are respectively analogous to VQ-VAE-2 and RQ-VAE, by proposing novel top-down layers (injected and residual top-down layers). In other words, our HQ-VAE encompasses two distinct model types, 1) a hierarchical structure, as in VQ-VAE-2, and 2) a residual quantization, as in RQ-VAE, within a unified formulation.
>
> (3) The hybrid model described in Appendices C and D.3 provides a recipe for hierarchical discrete representation modeling beyond SQ-VAE-2 and RSQ-VAE. It includes the construction of hierarchical latent structures and the derivation of objective functions for model learning.
>
> HQ-VAE offers the following advantages over the existing VQ-VAE models.
>
> (i) The self-annealing mechanism originating from the SQ-based formulation mitigates layer collapse. One can achieve better reconstruction accuracy than RQ-VAE and VQ-VAE-2 by training the counterparts in HQ-VAE. Furthermore, this property facilitates the training of a larger number of layers, as demonstrated in Figure 14.
>
> (ii) In VQ-VAE, certain hyperparameters (e.g., $\beta$ in the objective function, those for the EMA update and codebook reset) are introduced per layer.  Consequently, the number of hyperparameters increases as the number of layers grows. In contrast, HQ-VAE does not encounter this issue as there is no need to introduce any hyperparameter per layer.
>
> (iii) Constructing hierarchical models for information disentanglement is an interesting future direction.

---

> ### Author Response · Authors · 2024-02-16
> **Author reply (2/2)**
>
> **Requested Changes**
> ---
> > 1. What are differences between your approach to learn latent representations and VQ-VAE-2?
>
> Thank you for bringing up this question. SQ-VAE-2, an instance of HQ-VAE, is comparable to VQ-VAE-2 as they share the same architectural structure. The differences between them lie in their training strategy, specifically in their (1) quantization ways and (2) objective functions.
>
> (1) SQ-VAE-2 adopts a trainable stochastic quantization scheme, whereas VQ-VAE-2 quantizes encoded vectors deterministically into their nearest neighbors.
>
> (2) SQ-VAE-2 is trained by maximizing the ELBO in the variational Bayes framework. In contrast, VQ-VAE-2 relies on a manually-designed objective function, which involves the need for extra hyperparameter tuning.
>
> Thanks to the ELBO optimization with the stochastic quantization scheme, SQ-VAE-2 benefits from the self-annealing effect, mitigating layer collapse and resulting in better reconstruction accuracy.
>
> ---
> > 2-1. Section 4.3 needs to be improved for more clarification. What is $H_{\phi}(x)$?
>
> Thank you for your suggestion. Regarding the comment, we believe that associating $\mathbf{H}_{\phi}(x)$ with the notations in Section 3 would enhance the readability of our manuscript. In Section 3, we denoted the features generated by the bottom-up path as $\mathbf{H}\_{\mathbf{\phi}}^r$ with $r\in[R]$. In RSQ-VAE, since it includes only the residual layers in the top-down path, therefore dealing only with one resolution, we set $R=1$. For notational simplicity, we denoted $\mathbf{H}\_{\mathbf{\phi}}^1$ as $\mathbf{H}\_{\mathbf{\phi}}$, thus omitting the superscript. We will clarify the notation of $\mathbf{H}\_{\mathbf{\phi}}$.
>
> ---
> > 2-2. How can you do $\sum_{i=1}^lZ\_i$ because they have different dimensions?
>
> We appreciate the question, as it provides an important perspective on how to improve the manuscript. We would like to clarify the dimensionality regarding top-down layers. Basically, the dimension of $\mathbf{Z}\_l$ is the same as that of $\mathbf{H}\_{\mathbf{\phi}}^{r(l)}$. This means that the residual top-down layer at the $l$th layer maintains the same dimensionality for the latent $\mathbf{Z}\_l$ as it is for the previous layer (i.e., $\mathbf{Z}\_{l-1}$). Since the top-down path in RSQ-VAE consists only of residual top-down layers, all the latent $(\mathbf{Z}\_l)_{l=1}^L$ have the same dimensions. Furthermore, the residual top-down layer is designed to keep the dimensionality to conduct residual quantization, as in Equation (9). We will add explanations on the dimensionality in the revised manuscript.
>
> ---
> > 3. How to compute the bottom-up path $\mathbf{H}\_{\phi}^{r(l)}$? Is it the same as Stage 1: Learning Hierarchical Latent Codes in VQ-VAE?
>
> $\mathbf{H}\_{\mathbf{\phi}}^{r(l)}$, which is a part of the autoencoder, is computed using the standard feedforward calculation of the architecture. The autoencoding path (i.e., $\mathbf{H}\_{\mathbf{\phi}}^{r(l)}$ and $\mathbf{G}\_{\mathbf{\phi}}^{l}$) are jointly trained with the codebooks ($(\mathbf{B}_l)\_{l=1}^L$) in a similar manner to the stage 1 in VQ-VAE. Please refer to our responses to “Weakness” and “Requested Change 1” sections for the differences between our HQ-VAEs and the VQ-VAE variants.
>
> ---
> > 4. The authors should show the generated images for Table 3?
>
> Thank you for your comment. We acknowledge that the current manuscript lacks references to the generated images in the body. The generated images for Tables 3 and 4 can be found in Figures 15 and 16 in the appendix, respectively. We will include additional sentences in Section 5.4 to direct readers to the figures.

---

> ### Comment · Reviewer_bvQe · 2024-03-05
> **Thanks for your feedback**
>
> Thanks for answering my questions. They are all clear now.

---

> > ### Author Response · Authors · 2024-03-06
> > **Author reply**
> >
> > Thank you for your reply. We are glad to hear that our responses and revisions are valid for addressing your questions.

---

### Author Response · Authors · 2024-02-19
**To all reviewers**

First of all, we would like to express our deep appreciation to all the reviewers who have dedicated their time and expertise to reviewing our manuscript. In response to your comments, we have made the following changes to our manuscript highlighted in red. The summary of the modifications is as follows. **RC** and **W** stand for “Requested Changes” and “Weakness”, respectively.

1. We reorganized Section 2 and added a section (Appendix A) comparing SQ-VAE with other stochastic discrete models. **[Reviewer dLpy: RC]**, **[Reviewer MV41: RC3]**

2. We corrected/clarified the phrases, sentences, and formulas suggested by Reviewer MV41. **[Reviewer MV41: RC1, RC2, RC4, RC6, RC7, RC8, RC9]**

3. We provide further explanation in the caption of Figure 1 to help readers better understand our proposal. In particular, we added explanations of *top-down* layers in terms of dimensionality. **[Reviewer MV41: RC-5]**, **[Reviewer bvQe: RC2-2, RC3]**

4. We made a change to the figure to include the path from $\mathbf{x}$ to $\mathbf{H}\_{\mathbf{\phi}}^l(\mathbf{x})$ in “For posterior”, which is highlighted with red arrows. **[Reviewer bvQe: RC3]**

5. We clarified our intention in the remark right after Section 4.3.2. **[Reviewer MV41: RC10]**

6. We included a sentence in Section 5 to explain how we used the validation and test sets, and we also included the sizes of the datasets in Appendix E. **[Reviewer MV41: W2-1]**

7. We added experimental results to show the sensitivity of the models in terms of batch size in Figure 8. **[Reviewer MV41: W2-2]**

8. We added Appendix E to explain the approximation using the Gumbel-softmax in HQ-VAE. **[Reviewer MV41: RC11]**

9. We put sentences briefly explaining the differences between SQ-VAE-2 (RSQ-VAE) and VQ-VAE-2 (RQ-VAE) in the first paragraph of Section 4. **[Reviewer bvQe: RC1]**

10. We clarified the notations regarding $\mathbf{H}\_{\mathbf{\phi}}(\mathbf{x})$ more in Section 4.3. **[Reviewer bvQe: RC2-1]**

11. We put sentences in Section 5.4 to refer to the figures showing generated images from our models. **[Reviewer bvQe: RC4]**

---

### Decision · Action_Editor_RNuU · 2024-03-19

**Recommendation:** Accept as is

**Comment:**

The paper extends the vector quantization VAE to a hierarchical setting.

The reviewers note that the novelty of the paper is limited but the all agree that the paper fills an important gap in the literature and the paper after revisions is of very good quality.

**Audience:**

Yes.

**Claims And Evidence:**

Yes.